# ACTIVE EVALUATION ACQUISITION FOR EFFICIENT LLM BENCHMARKING

## ABSTRACT

As large language models (LLMs) become increasingly versatile, numerous large scale benchmarks have been developed to thoroughly assess their capabilities. These benchmarks typically consist of diverse datasets and prompts to evaluate different aspects of LLM performance. However, comprehensive evaluations on hundreds or thousands of prompts incur tremendous costs in terms of computation, money, and time. In this work, we investigate strategies to improve evaluation efficiency by selecting a subset of examples from each benchmark using a learned policy. Our approach models the dependencies across test examples, allowing accurate prediction of the evaluation outcomes for the remaining examples based on the outcomes of the selected ones. Consequently, we only need to acquire the actual evaluation outcomes for the selected subset. We rigorously explore various subset selection policies and introduce a novel RL-based policy that leverages the captured dependencies. Empirical results demonstrate that our approach significantly reduces the number of evaluation prompts required while maintaining accurate performance estimates compared to previous methods.

## 1 INTRODUCTION

By scaling up parameters and pretraining data, large language models (LLMs) have demonstrated remarkable abilities to solve various tasks. To reliably evaluate these capabilities and compare different models, modern LLM benchmarks typically employ a comprehensive set of examples that focus on different aspects of performance. For instance, HELM (Liang et al., 2022), a widely used LLM benchmark covering diverse task families, includes 42 scenarios and approximately 600,000 queries. However, the comprehensiveness of these benchmarks inevitably incurs significant evaluation costs. For example, evaluating on the HELM benchmark requires 4,200 GPU hours for a 176B BLOOM model and $9,337 for text-davinci-002 API calls (Liang et al., 2022). Furthermore, these substantial costs hamper development at both the modeling and inference stages, preventing frequent evaluation during the former and extensive hyperparameter tuning – such as decoding and prompting strategies – during the latter.

In this work, we aim to improve evaluation efficiency by reducing the number of evaluation prompts. We first observe that evaluation prompts are highly correlated, meaning that a model's (in)correct prediction on a certain prompt is likely to correspond with (in)correct predictions on related prompts. To leverage this, we build a model to formally capture the dependencies across prompts. This model can predict evaluation scores based on observed scores from a subset of prompts. Given these dependencies, our goal becomes identifying the minimal subset of prompts that can accurately recover the evaluation scores for the remaining prompts.

Instead of using a fixed subset of prompts across all models, we propose selecting a unique subset of prompts for each model to evaluate its performance more efficiently. The goal of our active evaluation acquisition (AEA) approach is to find the most informative subset of prompts for each model, allowing us to predict performance on the remaining prompts. The key insight is that models may have varying strengths; for example, one model may excel in arithmetic reasoning while another shows stronger commonsense reasoning. Tailoring the subset of prompts for each model ensures a more accurate and targeted evaluation of its capabilities. Furthermore, our dynamic acquisition process allows us to adapt in real time as evaluation scores are gathered. As the model's performance on initial prompts is observed, the system adjusts subsequent prompt selections to better explore areas

of uncertainty or confirm early findings. This iterative approach not only enhances the accuracy of performance estimation but also reduces redundancy by avoiding prompts that are likely to yield predictable results, thereby saving computational resources and time. Importantly, the final evaluation score is derived from both the acquired scores on selected prompts and predicted scores on the remaining prompts, ensuring comparability across models is maintained.

Our contributions are as follows: 1) We tackle LLM evaluation efficiency through dependency modeling and subset selection, connecting LLM evaluation with the extensive literature on subset selection. 2) We design a generative model that captures dependencies across evaluation prompts and handles mixed-type evaluation scores, including both discrete and real-valued scores. 3) We thoroughly test existing subset selection algorithms on several popular LLM evaluation benchmarks, including MMLU (Hendrycks et al., 2020), HELM (Liang et al., 2022), HuggingFace Open LLM Leaderboard (Beeching et al., 2023), AlpaceEval (Li et al., 2023), and Chatbot Arena (Zheng et al., 2024). 4) We develop several new subset selection policies based on the dependency model and demonstrate their superiority over existing methods, with our RL-based acquisition policy achieving the best performance using the lowest acquisition budget. 5) We propose and investigate the cold-start problem, where new prompts are added to a benchmark without prior evaluation scores for any model, and extend our RL-based policy to deal with the situation effectively.

## 2 METHOD

### 2.1 PROBLEM FORMULATION

Consider a benchmark $X$ with $N$ prompts, $X = \{x_n\}_{n=1}^N$. Note that a benchmark may contain multiple datasets. Evaluating a model $m$ on this benchmark generates evaluation scores $Y_m = \{y_{mn}\}_{n=1}^N$. A leaderboard for this benchmark contains evaluation scores for $M$ models, denoted as $\{Y_m\}_{m=1}^M$. For a new model $m'$ to be evaluated, our AEA framework will acquire a subset of the evaluation scores $Y_{m'}^{(o)} = \{y_{m'o} : o \subseteq \{1, \ldots, N\}\}$, and the evaluation scores for the remaining prompts $Y_{m'}^{(u)} = \{y_{m'u}; u = \{1, \ldots, N\} \setminus o\}$ will be predicted. The key of our AEA framework is to capture the dependencies over prompts so that the predicted evaluation scores are accurate. We explicitly model the dependencies by learning the conditional distribution $p(Y_m^{(u)} \mid Y_m^{(o)}, X)$. Since the set of prompts to acquire their scores is not predefined, we must estimate $p(Y_m^{(u)} \mid Y_m^{(o)}, X)$ for all possible subsets $u$ and $o$.

Given the generative model between subsets of evaluation scores and a fixed budget $K$, our goal for AEA is to find an optimal subset $o^* \subseteq \{1, \ldots, N\}$, where $|o^*| = K$, such that the predicted scores on the remaining prompts are accurate, i.e.,

$$o^* = \arg\max_{o \in \mathbb{P}([N], K)} p(Y_{m'}^{(u)} \mid Y_{m'}^{(o)}, X), \tag{1}$$

where $\mathbb{P}([N], K)$ represents all subsets of $\{1, \ldots, N\}$ with cardinality $K$. Note that the optimal subset $o^*$ could be different for each model; however, for notation simplicity, we omit the subscript $m'$. It is worth noting that the objective in equation 1 cannot be directly optimized since the values $Y_{m'}^{(u)}$ for a test model $m'$ are unknown before we actually acquire the evaluation scores on those prompts. Additionally, equation 1 bares similarity to the objective of active learning (Ren et al., 2021), but our goal differs in that we aim to achieve more accurate evaluations rather than training a better model. We will later assess several acquisition policies inspired by active learning algorithms.

### 2.2 MODELING DEPENDENCIES VIA NEURAL PROCESSES

In this section, we aim to capture the dependencies across evaluation prompts by modeling the conditional distribution $p(Y_m^{(u)} \mid Y_m^{(o)}, X)$. We represent the relationship between the prompts and their evaluation scores as a stochastic process $F : \mathcal{X} \to \mathcal{Y}$, where $\mathcal{X}$ and $\mathcal{Y}$ denote the spaces of prompts and their corresponding evaluation scores, respectively. Evaluating on the benchmark $X$ can be interpreted as observing finite-dimensional marginal distributions of this stochastic process. Specifically, the evaluation scores $Y_m$ represent the function values $\{f_m(x_n)\}_{n=1}^N$ for a particular function instantiation, $f_m$, sampled from the distribution of functions.

Neural Processes (NPs) (Garnelo et al., 2018b;a; Kim et al., 2019) provide a flexible and scalable approach to modeling such stochastic processes. They combine the strengths of neural networks and Gaussian Processes to predict outputs for new inputs by conditioning on a set of context points. Specifically, the function $f_m$ is implicitly parameterized by a latent vector $z_m$, and the generative model then follows

$$p(Y_m^{(u)} \mid Y_m^{(o)}, X) = \int p(z_m \mid Y_m^{(o)}, X) p(Y_m^{(u)} \mid z_m, Y_m^{(o)}, X) dz_m. \tag{2}$$

Since the integration is over a high dimensional latent space, we instead optimize the evidence lower bound (ELBO) following variational autoencoder (VAE) (Kingma & Welling, 2013)

$$\log p(Y_m^{(u)} \mid Y_m^{(o)}, X) \geq \mathbb{E}_{q(z_m \mid Y_m^{(u)}, Y_m^{(o)}, X)} \left[ \log \frac{p(Y_m^{(u)} \mid z_m, Y_m^{(o)}, X) p(z_m \mid Y_m^{(o)}, X)}{q(z_m \mid Y_m^{(u)}, Y_m^{(o)}, X)} \right], \tag{3}$$

where $q(z_m \mid Y_m^{(u)}, Y_m^{(o)}, X)$ and $p(z_m \mid Y_m^{(o)}, X)$ represent the posterior and prior distributions over the latent variable, respectively.

To ensure that our model represents a valid stochastic process, we adhere to the conditions stated by the Kolmogorov Extension Theorem (Oksendal, 2013): (finite) exchangeability and consistency. The exchangeability condition requires that the joint distribution $p(Y_m)$ remain unchanged under permutations of its elements. In practice, we can satisfy this condition by using permutation invariant networks to parameterize both the prior and posterior distributions. The consistency demands that if we marginalize out part of $Y_m$, the resulting marginal distribution is the same as that defined on the original prompt $x_n$. This condition is met when the approximate posterior equals the true posterior. In practice, we achieve this by training the model with a sufficient amount of data from diverse model evaluations so that the lower bound approaches the actual likelihood.

**Implementation** In order to handle textual prompts, we utilize a pretrained embedding model to represent each prompt as a $\mathbb{R}^d$ vector. During training, since the entire set $Y_m$ might be too large to fit into memory, we randomly sample two non-overlapping subsets from each model as $Y_m^{(o)}$ and $Y_m^{(u)}$, respectively. The prior and posterior distributions share the same network, but take different inputs. The prior takes in a set of x-y pairs from $Y_m^{(o)}$, i.e., $\{(x_o, y_{mo}) : o \subseteq \{1, \ldots, N\}\}$, while the posterior takes in a set of x-y pairs from both $Y_m^{(o)}$ and $Y_m^{(u)}$. Following the Attentive Neural Process (Kim et al., 2019), we implement the prior/posterior network using self-attention blocks to better capture the dependencies across set elements. To reduce memory usage, we use Set Transformer architecture (Lee et al., 2018), where each set element attends to a small set of learnable induced points instead of attending to all other elements directly. The decoder network $p(Y_m^{(u)} \mid z_m, Y_m^{(o)}, X)$ employs cross-attention, allowing each unobserved prompt to attend to the relevant observed prompts. According to De Finetti's Theorem (De Finetti, 1929), the likelihood over set elements $Y_m^{(u)}$ can be conditionally independent conditioned on the latent variable $z_m$. However, we still use a Set Transformer (Lee et al., 2018) to better capture the dependencies. Please refer to Appendix A for details of the model architecture.

## 2.3 Evaluation Acquisition Policy

Given the generative model across subsets of evaluation prompts, we now develop acquisition policies to select an optimal subset of prompts for acquiring their true evaluation scores, while the remaining scores will be predicted by the conditional $p(Y_m^{(u)} \mid Y_m^{(o)}, X)$.

### 2.3.1 Random Policy

A random acquisition policy selects a subset of size $K$ at random to acquire the evaluation scores. In this work, we consider two variants: **Uniform Sampling** and **Stratified Random Sampling** (Perlitz et al., 2023). Uniform Sampling selects $K$ prompts uniformly from $X$, while Stratified Random Sampling considers the size of different datasets and ensures each dataset is equally represented. The stratified sampling has been verified effective on HELM benchmark (Perlitz et al., 2023).

---

**Algorithm 1** Active Evaluation Acquisition

---

**Require:** Acquisition budget $K$, a model $m$ to be evaluated, Neural Process $p$

1: $o = \emptyset, Y_m^{(o)} = \emptyset, u = \{1, \ldots, N\}$
2: **while** $|o| < K$ **do**
3:     Select prompt $i$ according to equation 5, equation 6, or equation 7
4:     Run evaluation for model $m$ on prompt $i$ to get the evaluation score $Y_m^{(i)}$
5:     $o = o \cup \{i\}, Y_m^{(o)} = Y_m^{(o)} \cup \{Y_m^{(i)}\}, u = u \setminus \{i\}$
6: **end while**
7: Predict the evaluation scores for the remaining prompts $Y_m^{(u)} \sim p(Y_m^{(u)} \mid Y_m^{(o)}, X)$

---

### 2.3.2 STATIC POLICY

A static acquisition policy determines the set of prompts to be evaluated beforehand, and each model to be evaluated acquires the evaluation scores on the same set of prompts. We assess the following two types of static policies.

**Clustering** Given the embedding for each prompt, we group them into $K$ cluster, then we select one prompt in each cluster that is closest to the cluster centroid. We denote this approach as **Clustering-Embed**. Instead of using the pretrained sentence embedding, we can use the learned embedding from an Item Response Theory (IRT) model (Hambleton & Swaminathan, 2013; Embretson & Reise, 2013), which represents the difficulty and discriminability of each prompt. The **Clustering-IRT** method, proposed in (Polo et al., 2024), has been successfully applied on several public LLM benchmarks. Inspired by (Vivek et al., 2023), which selects representative examples by clustering based on prediction confidence, the **Clustering-Score** method groups the prompts based on their evaluation scores on the training set. That is, each prompt $x_n$ is represented by a vector of evaluation scores, with the size of the vector corresponding to the number of evaluated models in the training set. The Clustering-Score method has been used as a baseline in (Polo et al., 2024).

**Combinatorial Optimization** Given the model $p(Y_m^{(u)} \mid Y_m^{(o)}, X)$, a static acquisition policy can be derived by searching over the training set to find the optimal subset of prompts that gives the most accurate prediction of the remaining prompts. This is a typical combinatorial optimization problem, which is NP-Hard. Here, we employ a sequential approach that selects one prompt at a time until $K$ prompts are selected. Starting from an empty set $o = \emptyset$, the next prompt $i \in u := \{1, \ldots, N\} \setminus o$ is chosen to minimize the prediction error over the training set, i.e.,

$$i = \underset{i' \in u}{\arg\min} \, \mathbb{E}_{Y_m \sim p_{\mathcal{D}}} \mathbb{E}_{\hat{Y}_m^{(u')} \sim p(Y_m^{(u')} | Y_m^{(o')}, X)} \| \hat{Y}_m^{(u')} - Y_m^{(u')} \|^2, \tag{4}$$

where $o' = o \cup \{i'\}$ and $u' = u \setminus \{i'\}$. We estimate the expectation by Monte Carlo sampling. For notation simplicity, the above equation computes the mean squared error on prompts $u'$; however, in practice, different datasets may use different metrics. Additionally, these differences may be weighted depending on the dataset size. Please refer to Algorithm 2 in Appendix for pseudo-code of the selection process. Note that this approach has a complexity of $O(KMN)$, which could be prohibitive when the benchmark is large.

### 2.3.3 DYNAMIC POLICY

Instead of acquiring the same set of evaluation scores for each model, we propose dynamically acquiring adaptive subsets for different models, a method we term Active Evaluation Acquisition (AEA). This approach tailors the selection of prompts to each model's specific strengths and weaknesses, providing a more accurate and efficient evaluation. Dynamic acquisition sequentially acquires evaluation scores and simultaneously refines the uncertainty of predictions, enabling real-time adaptation based on observed performance. AEA reduces redundancy by avoiding predictable evaluations and focusing resources on the most informative prompts. Please refer to Algorithm 1 for pseudo-code of the active acquisition process.

**Uncertainty Sampling** Inspired by uncertainty sampling method widely used in active learning literature(Ren et al., 2021; Yang et al., 2015; Raj & Bach, 2022), where the most uncertainty data point under the current predictor is chosen to query its label, we select the next prompt to be evaluated

based on the uncertainty of $p(Y_m^{(i)} \mid Y_m^{(o)}, X)$. Here, $o$ contains the evaluated prompts so far, and $i \in u$ is one of the candidate prompts to be selected. We choose the prompt with the highest entropy:

$$i = \arg\max_{i \in u} H(Y_m^{(i)} \mid Y_m^{(o)}, X). \tag{5}$$

In practice, we estimate the entropy by sampling multiple times and computing the sample variance.

**Information Gain** Given the latent variable based neural process model (equation 2), where the latent variable essentially parameterizes the stochastic process, a straight-forward acquisition policy is to select the prompt that provides the most information about the latent variable $z_m$. We use the conditional mutual information to measure the amount of information:

$$
\begin{aligned}
i &= \arg\max_{i \in u} I(Y_m^{(i)}; z_m \mid Y_m^{(o)}, X) \\
&= \arg\max_{i \in u} \left[ H(z_m \mid Y_m^{(o)}, X) - \mathbb{E}_{\hat{Y}_m^{(i)} \sim p(Y_m^{(i)} \mid Y_m^{(o)}, X)} H(z_m \mid \hat{Y}_m^{(i)}, Y_m^{(o)}, X) \right] \\
&= \arg\min_{i \in u} \mathbb{E}_{\hat{Y}_m^{(i)} \sim p(Y_m^{(i)} \mid Y_m^{(o)}, X)} H(z_m \mid \hat{Y}_m^{(i)}, Y_m^{(o)}, X).
\end{aligned} \tag{6}
$$

The third equation follows because the observed set $o$ is the same for any candidate $i \in u$. The expectation is again estimated by Monte Carlo sampling. Note that the entropy is estimated based on predicted $Y_m^{(i)}$ rather than the true evaluation score as the true score is unknown before acquisition. At each acquisition step, the entropy must be estimated for each candidate prompt $i \in u$. Therefore, the total complexity is $O(KN)$, which could be prohibitive for large benchmarks.

**Reinforcement Learning** The active acquisition process can be formulated as a Markov decision process (MDP), where the state consists of the currently evaluated prompts and their scores, and the action space contains the remaining prompts to be evaluated. To solve the MDP, a reinforcement learning agent sequentially acquires new evaluation scores based on the current state. After acquiring evaluation score for prompt $i$, the current state transitions to a new state as follows: $o \xrightarrow{i} o \cup \{i\}, Y_m^{(o)} \xrightarrow{i} Y_m^{(o)} \cup \{Y_m^{(i)}\}$. When the agent acquires evaluation scores for $K$ prompts, the acquisition process terminates, and the agent receives a reward based on the prediction accuracy for the remaining prompts. Note that the reward is only required during training when we have access to all the evaluation scores, allowing us to compute the actual prediction accuracy. During testing, the next prompt to be evaluated will be directly selected by the policy:

$$i = \arg\max_{i \in u} P(i \mid Y_m^{(o)}, X). \tag{7}$$

Since the policy network has constant computational cost, the total complexity of the acquisition process remains $O(K)$ regardless of the benchmark size.

In the above MDP definition, the reward is received only at the end of the acquisition process by predicting the unobserved evaluation scores. This setup poses a typical temporal credit assignment problem, which complicates the learning of an effective agent, especially when the trajectory is long (Minsky, 1961; Sutton, 1988). To address this issue, we propose providing intermediate rewards for each acquisition action $i$. Specifically, after acquiring the evaluation score for prompt $i$, the improvement in prediction accuracy per unobserved prompt is used as the intermediate reward, i.e.,

$$r(o, i) = \frac{\mathbb{E}_{\hat{Y}_m^{(u)} \sim p(Y_m^{(u)} \mid Y_m^{(o)}, X)} \|\hat{Y}_m^{(u)} - Y_m^{(u)}\|^2}{|u|} - \frac{\mathbb{E}_{\hat{Y}_m^{(u')} \sim p(Y_m^{(u')} \mid Y_m^{(o')}, X)} \|\hat{Y}_m^{(u')} - Y_m^{(u')}\|^2}{|u'|}, \tag{8}$$

where $o' = o \cup \{i\}$ and $u' = u \setminus \{i\}$. The intermediate reward provides immediate feedback for each acquisition action during the acquisition process, facilitating more effective learning. Note that the intermediate reward follows the potential function structure (Ng et al., 1999), therefore, it will not change the optimal policy.

In addition to providing intermediate rewards, we propose using the neural process to assist the agent with auxiliary information. Specifically, the neural process can predict the evaluation scores for unobserved prompts based on the observed scores in the current state. By sampling multiple times, the neural process can inform the agent about the uncertainties of these unobserved scores. The predicted scores and their uncertainties on the unobserved prompts allow the agent to anticipate future states and guide its exploration. For instance, if the neural process is very confident about the score of a currently unobserved prompt, then acquiring its real score would be redundant. The auxiliary information helps the agent make more informed decisions about which prompts to evaluate next, improving the efficiency and accuracy of the active acquisition process.

## 2.4 COLD START PROBLEM

So far, we have considered scenarios where the set of prompts is fixed for each benchmark. However, as language models advance, new capabilities may emerge that need to be assessed. Therefore, it is crucial to address the cold start problem, where the benchmark must be expanded with new prompts for which no evaluation scores are initially available for any model.

Expanding the benchmark with new prompts introduces several challenges. Firstly, predicting evaluation scores on these new prompts is difficult for a neural process trained on previously observed scores. Secondly, determining which new prompts to select for the expanded benchmark is challenging, as these new prompts may introduce capabilities or areas not well-represented in the original set, making it hard to gauge their relevance and difficulty relative to the existing prompts.

To help the neural process model generalize to new prompts, we introduce a semi-supervised training procedure where the new prompts are treated as unlabeled data. We found that simple pseudo-labeling approaches (Lee et al., 2013; Xie et al., 2020; Du et al., 2020) work well. Specifically, we add the new prompts and their predicted evaluation scores into the training process if the uncertainties of the predicted evaluation scores are below a predefined threshold. In our preliminary experiments, we also tested several regularization approaches, such as entropy minimization (Grandvalet & Bengio, 2004) and consistency regularization (Tarvainen & Valpola, 2017), which are commonly used for semi-supervised learning, but they consistently underperformed compared to pseudo-labeling. A systematic exploration of semi-supervised techniques for neural process training is beyond the scope of this paper and will be left for future work.

Static acquisition policies are suboptimal for the cold start problem because they rely on the available evaluation scores to determine the set of prompts to be evaluated, meaning the evaluation scores on new prompts will never be acquired. One exception is Clustering-Embed method, where the clustering is based solely on the embedding of the new prompts rather than their evaluation scores. In contrast, dynamic acquisition policies are better suited to handle the cold start problem due to their adaptive nature. However, since the RL-based acquisition policy is trained to acquire evaluation scores on the existing prompts, it requires the acquisition policy to generalize to new prompts.

To enable the policy network to generalize to new actions, we design it to incorporate the action representations into its inputs. Specifically, at each acquisition step, in addition to the acquired scores $Y_m^{(o)}$, the policy network $h$ also takes in the representations of the available actions, where the representations are shared with the neural process model. The output of the policy network is a vector with the same dimensionality as the action representations. The probability of selecting a particular prompt $i$ is proportional to the inner product of the output vector and action representations, i.e,

$$P(i \mid Y_m^{(o)}, X) = \frac{e^{a_i \cdot h(Y_m^{(o)}, X^{(o)}, \{a_i\}_{i \in u})}}{\sum_{i \in u} e^{a_i \cdot h(Y_m^{(o)}, X^{(o)}, \{a_i\}_{i \in u})}}, \tag{9}$$

where $\{a_i\}_{i \in u}$ denote the representations for the set of candidate prompts. A similar policy architecture design has been proposed before for sequential decision-making (Jain et al., 2020). Please refer to Appendix D for further details.

## 3 RELATED WORKS

**Active Learning** Active learning (Fu et al., 2013; Konyushkova et al., 2017; Yoo & Kweon, 2019) addresses the problem of having a learner select specific examples to query an oracle for their labels, with the goal of learning a better model using as few labeled examples as possible. In contrast, our proposed AEA framework focuses on evaluating a model with fewer examples to accurately predict the evaluation scores for the remaining examples.

**Active Testing** Active testing (Kossen et al., 2021) reduces the labeling cost by selectively choosing test points to label, ensuring sample-efficient model evaluation. While this aligns with the goal of efficient evaluation, our work specifically targets reducing the cost of running evaluations on a large number of prompts, rather than minimizing labeling costs.

**Efficient LLM Benchmarking** As LLMs continue to develop and scale, ongoing efforts aim to create benchmarks that comprehensively assess their capabilities. A notable trend in these benchmarks is their evolution from single-task assessments (Bowman et al., 2015; Rajpurkar et al., 2016)

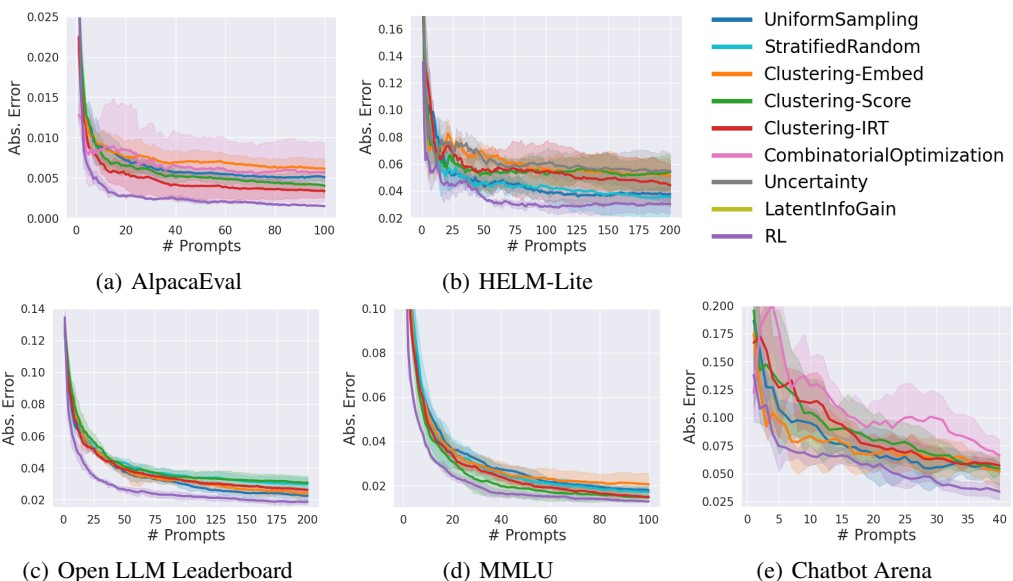

Figure 1: Experiment results on five popular LLM evaluation benchmarks, with shaded areas indicating the standard deviation over three runs.

to multi-task benchmarks (Wang et al., 2018; 2019), and ultimately to massively multi-task evaluations (Srivastava et al., 2022; Liang et al., 2022; Hendrycks et al., 2020). The ever-increasing evaluation cost has encouraged researchers to develop efficient evaluation approaches. BIG-bench Lite (Srivastava et al., 2022) and BIG-bench Hard (Suzgun et al., 2022) evaluate on a subset of BIG-bench tasks, and Ye et al. (2023) propose clustering BIG-bench tasks and selecting the examples that are closest to cluster centers. Perlitz et al. (2023) found that the model rankings on HELM can be accurately obtained by evaluating only a fraction of the examples. Vivek et al. (2023) propose clustering the evaluation examples based on the uncertainty of model predictions, while Polo et al. (2024) suggest clustering examples based on learned features from an IRT model. In this work, we comprehensively assess these methods and further propose actively selecting evaluation examples.

## 4 EXPERIMENTS

In this section, we assess various evaluation acquisition policies on several popular LLM benchmarks. We divide the available leaderboard scores into training and test splits. The training split is used to fit the neural processes model, capturing the dependencies across prompts. The acquisition policies are executed for each model in the test split to acquire the evaluation scores for a subset of prompts. The evaluation scores on the remaining prompts are predicted based on the corresponding neural process model. The final score for each benchmark is computed as a weighted average across datasets, and we report the absolute differences between the predicted scores and the actual scores. Please see Appendix E for details.

We conduct experiments on five popular LLM benchmarks: HuggingFace Open LLM Leaderboard (Beeching et al., 2023), MMLU (Hendrycks et al., 2020), HELM-Lite (Liang et al., 2022), AlpacaEval 2.0 (Li et al., 2023), and Chatbot Arena (Zheng et al., 2024). Detailed descriptions of these benchmarks can be found in Appendix E.

**Results** Figure 1 presents the main experimental results on 5 LLM benchmarks. We conduct experiments with 3 random seeds for each benchmark and plot the average performance and standard deviation throughout the acquisition process. Prompt embeddings are obtained using the SFR embedding model (Rui Meng, 2024). For the static clustering based policies, since the selected prompts do not have an inherent order, the acquisition process shuffles the selected prompts at random. For the AlpacaEval and Chatbot Arena benchmarks, stratified random sampling is equivalent to uniform sampling since there are only one dataset in each benchmark. Combinatorial optimization is too ex-

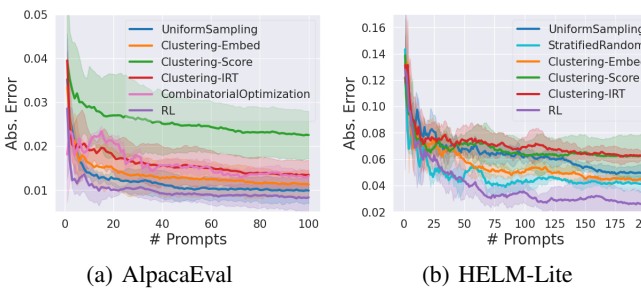 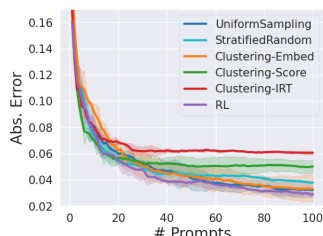

(a) AlpacaEval     (b) HELM-Lite

Figure 3: Evaluate the cold start problem on MMLU benchmark, where 15 subsets are left out as cold start prompts.

Figure 2: Evaluate the situation with model bias, where test models are from different model families compared to the training models.

pensive to run for HELM-Lite, HuggingFace Open LLM Leaderboard, and MMLU due to the large number of prompts. We found that uncertainty and information gain based policies consistently fail to explore the action space, leading to worse overall performance. To avoid cluttering the plots, results for uncertainty sampling and information gain based policies are moved to the appendix. Please refer to Appendix E for more analysis.

For all benchmarks, our proposed RL-based acquisition policy achieves the best performance with the lowest acquisition budget, demonstrating its superior ability to select informative prompts and accurately estimate benchmark performance. The stratified random sampling policy performs similarly to uniform sampling. Interestingly, the Clustering-Embed policy does not outperform the random selection, indicating that the similarity in prompt embedding does not always translate to the similarity in evaluation scores. Among the three clustering-based policies, none consistently outperforms the others. On AlpacaEval, HELM-Lite, and MMLU, the policies that utilize the evaluation scores (i.e., Claustering-Score and Clustering-IRT) perform better, while on the Open LLM Leaderboard and Chatbot Arena, Clustering-Embed perform better. The combinatorial optimization based policy does not perform well, even on the two small benchmarks where it is computationally feasible. We attribute this to a potential distribution shift between the models used for training and those used for testing, suggesting that the static policy optimized on training models does not generalize well to new models during testing.

Additionally, we compare our methods with tinybenchmarks (Polo et al., 2024). Given the selected subsets from tinybenchmarks, we predict the final benchmark performance using both the IRT models provided by tinybenchmark [1] and our neural process models. Conversely, we also evaluate our prompt selections using both IRT and NP models. Table 1 compares the prompt selections from our proposed RL policy with those from tinybenchmark. The original tinybenchmark select 600 prompts for Huggingface Open LLM Leaderboard, but in our comparison, we select 200 prompts to ensure a fair comparison with our RL policy. The results show that for both prompt selections, us-

Table 1: Comparison of our RL-based acquisition policy with TinyBenchmarks (TB) (Polo et al., 2024), using selected prompts to predict evaluation scores with either the IRT model from TB or our NP model. The metric is the absolute error in benchmark score estimation.

|  |  | IRT | NP |
|---|---|---|---|
| AlpacaEval (K=100) | TB | $0.027 \pm 0.002$ | $0.003 \pm 0.001$ |
|  | RL | $\mathbf{0.014 \pm 0.005}$ | $\mathbf{0.001 \pm 0.000}$ |
| MMLU (K=100) | TB | $\mathbf{0.022 \pm 0.000}$ | $0.016 \pm 0.000$ |
|  | RL | $0.028 \pm 0.002$ | $\mathbf{0.013 \pm 0.000}$ |
| Open LLM (K=200) | TB | $0.023 \pm 0.002$ | $0.022 \pm 0.004$ |
|  | RL | $\mathbf{0.019 \pm 0.001}$ | $\mathbf{0.018 \pm 0.001}$ |

ing NP produces better benchmark performance estimates, indicating that our neural process model better captures the dependencies and predicts the missing evaluation scores. Given a fixed prediction model (either IRT or NP), our RL-based acquisition policy achieves lower error compared to the prompt selections from tinybenchmark, demonstrating that our RL-based policy is more effective at selecting the informative prompts.

**Model Bias** An important aspect of efficient benchmarking strategies is robustness to model bias. To accurately evaluate future models, which may differ significantly from previously seen models, the

---

[1]https://github.com/felipemaiapolo/tinyBenchmarks/tree/main

Table 2: Comparson of the final benchmark performance estimation methods. **w/ pred** indicate the proposed method where the neural process is used to predict the missing evaluation scores. **w/o pred** indicates the baseline where final performance is an aggregation of the acquired evaluation scores.

| | | AlpacaEval (K=100) | HELM-Lite (K=200) | Open LLM (K=200) | MMLU (K=100) | Chatbot Arena (K=40) |
|---|---|---|---|---|---|---|
| Uniform | w/ pred | 0.005 ± 0.000 | 0.038 ± 0.005 | 0.022 ± 0.002 | 0.018 ± 0.001 | 0.052 ± 0.010 |
| | w/o pred | 0.012 ± 0.001 | 0.079 ± 0.008 | 0.043 ± 0.003 | 0.042 ± 0.003 | 0.036 ± 0.009 |
| S-Rand | w/ pred | - | 0.035 ± 0.012 | 0.030 ± 0.003 | 0.017 ± 0.002 | - |
| | w/o pred | - | 0.072 ± 0.012 | 0.023 ± 0.001 | 0.038 ± 0.001 | - |
| C-Embed | w/ pred | 0.006 ± 0.001 | 0.051 ± 0.010 | 0.024 ± 0.002 | 0.020 ± 0.004 | 0.052 ± 0.014 |
| | w/o pred | 0.023 ± 0.008 | 0.116 ± 0.003 | 0.029 ± 0.000 | 0.029 ± 0.000 | 0.032 ± 0.003 |
| C-Score | w/ pred | 0.004 ± 0.001 | 0.054 ± 0.010 | 0.031 ± 0.003 | 0.014 ± 0.002 | 0.054 ± 0.004 |
| | w/o pred | 0.141 ± 0.011 | 0.051 ± 0.017 | 0.086 ± 0.002 | 0.048 ± 0.002 | 0.037 ± 0.011 |
| C-IRT | w/ pred | 0.003 ± 0.001 | 0.044 ± 0.013 | 0.026 ± 0.001 | 0.015 ± 0.001 | 0.057 ± 0.002 |
| | w/o pred | 0.069 ± 0.003 | 0.060 ± 0.014 | 0.037 ± 0.003 | 0.041 ± 0.006 | 0.042 ± 0.003 |
| RL | w/ pred | 0.001 ± 0.000 | 0.030 ± 0.005 | 0.018 ± 0.001 | 0.013 ± 0.000 | 0.034 ± 0.006 |
| | w/o pred | 0.064 ± 0.006 | 0.081 ± 0.019 | 0.063 ± 0.018 | 0.050 ± 0.006 | 0.045 ± 0.007 |

strategy must accurately measure model capabilities based on the selected prompts. Our train-test splits based on date for MMLU and Open LLM Leaderboard potentially evaluate this situation since model performance tends to improve over time. To further evaluate the performance in the presence of model bias, we divide the models on the AlpacaEval and HELM-Lite leaderboards based on their organizations. For HELM-Lite, we use proprietary models, such as GPT-4 (Achiam et al., 2023) and Claude (Anthropic, 2024), for training and test on open-source models, such as LLaMA (Touvron et al., 2023) and Mistral (Jiang et al., 2023). For AlpacaEval, we do the opposite, using open-source models for training and proprietary models for testing.

Figure 2 presents the evaluation results on these two benchmarks with model bias. Firstly, static policies, especially Clustering-Score and Clustering-IRT that depend on evaluation scores from the training models, do not perform well. Secondly, although random policies do not suffer from model bias, they cannot leverage dependencies across prompts, leading to lower overall performance. In contrast, our RL-based dynamic acquisition policy can effectively exploit the dependencies across prompts even for models that is significantly different from the models it has seen before. However, we do notice that the existence of model bias makes the problem harder to solve. Compared to Fig. 1 on the same benchmark, even for our RL-based policy, it takes more acquisitions to achieve the same level of errors as in situations where no model bias exists. In practice, a continual learning framework, where the neural process model and the acquisition policies are jointly adapted to the newly added models, might be necessary. We leave this for future works.

**Cold Start Problem** To evaluate the cold start scenario, we create a synthetic benchmark using MMLU by designating 15 subsets as cold start prompts. During the training of the neural process model and the acquisition policies, the evaluation scores on these 15 subsets are not available. Although the evaluation scores are missing, we assume the prompts themselves are given, allowing random policies and Clustering-Embed static policy to be evaluated without any modifications. However, the Clustering-Score and Clustering-IRT policies will never acquire evaluation scores for these 15 subsets since these policies require access to the evaluation scores to determine whether a prompt will be acquired or not. On the other hand, dynamic acquisition policies can easily adapt to the cold start setting, as they acquire evaluation scores sequentially and actively.

Figure 3 presents the results on the synthetic cold start MMLU benchmark. The performance is evaluated over all 57 subsets during testing. As expected, the Clustering-Score and Clustering-IRT policies do not perform well in the cold start setting because the evaluation scores on the 15 left-out subsets are never acquired. The Clustering-Embed policy performs better than the other clustering based policies as it can select the cold start prompts by clustering based on their embeddings. The RL-based acquisition policy again achieves the best performance estimation. However, it is worth noting that the final estimated benchmark performance is not as accurate as in the fully observed setting (Fig. 1), indicating potential areas for future improvement to narrow the gap.

### 4.1 ABLATION STUDIES

**Prediction Model** In the main experimental results, we run the acquisition policy to select a subset of prompts for acquiring their actual evaluation scores and then use a neural process model to predict

Table 3: Comparison of different prompt embeddings.

| | | Uniform | C-Embed | C-Score | C-IRT | RL |
|---|---|---|---|---|---|---|
| AlpacaEval | SFR (4096) | $0.005 \pm 0.000$ | $0.006 \pm 0.001$ | $0.004 \pm 0.001$ | $0.003 \pm 0.001$ | $0.001 \pm 0.000$ |
| | E5 (4096) | $0.005 \pm 0.000$ | $0.006 \pm 0.001$ | $0.007 \pm 0.005$ | $0.004 \pm 0.001$ | $0.001 \pm 0.000$ |
| | BGE-large (1024) | $0.006 \pm 0.001$ | $0.006 \pm 0.002$ | $0.009 \pm 0.002$ | $0.007 \pm 0.001$ | $0.002 \pm 0.000$ |
| | BGE-small (384) | $0.009 \pm 0.002$ | $0.009 \pm 0.002$ | $0.038 \pm 0.031$ | $0.015 \pm 0.010$ | $0.005 \pm 0.002$ |
| MMLU | SFR (4096) | $0.018 \pm 0.001$ | $0.020 \pm 0.004$ | $0.014 \pm 0.002$ | $0.015 \pm 0.001$ | $0.013 \pm 0.000$ |
| | E5 (4096) | $0.018 \pm 0.002$ | $0.018 \pm 0.002$ | $0.014 \pm 0.003$ | $0.016 \pm 0.002$ | $0.014 \pm 0.003$ |
| | BGE-large (1024) | $0.029 \pm 0.003$ | $0.028 \pm 0.005$ | $0.027 \pm 0.005$ | $0.023 \pm 0.006$ | $0.023 \pm 0.003$ |
| | BGE-small (384) | $0.028 \pm 0.003$ | $0.023 \pm 0.006$ | $0.022 \pm 0.005$ | $0.022 \pm 0.005$ | $0.022 \pm 0.002$ |

Table 4: Contributions of auxiliary information and intermediate reward for our RL-based policy.

| | AlpacaEval | MMLU | Open LLM |
|---|---|---|---|
| PPO | $0.004 \pm 0.002$ | $0.017 \pm 0.004$ | $0.033 \pm 0.010$ |
| +auxiliary_info | $0.003 \pm 0.001$ | $0.016 \pm 0.003$ | $0.029 \pm 0.005$ |
| +interm_reward | $\mathbf{0.001 \pm 0.000}$ | $\mathbf{0.013 \pm 0.000}$ | $\mathbf{0.018 \pm 0.001}$ |

the scores for the remaining prompts. However, an alternative method to estimate benchmark performance is to directly aggregate the acquired evaluation scores without relying on another model for prediction. The aggregation computes performance per dataset first and then averages across datasets. Table 2 compares these two estimation methods. The results show that the prediction model generally provides better benchmark performance estimation.

**Prompt Embedding** Our approach utilizes a sentence embedding model to extract representations for the prompts. These representations are used both to train the neural process model and to build the acquisition policies. For the main results, we use the SFR embedding model (`Salesforce/SFR-Embedding-Mistral`) (Rui Meng, 2024) to extract prompt representations. In Table 3, we present results using several other embedding models: E5, BGE-large, and BGE-small, corresponding to `intfloat/e5-mistral-7b-instruct` (Wang et al., 2023), `BAAI/bge-large-en-v1.5` (Xiao et al., 2023), and `BAAI/bge-small-en-v1.5` (Xiao et al., 2023), respectively. The results indicate that performance generally improves with more powerful embedding models that better distinguish text inputs [2] Thus, utilizing more powerful embedding models is an important future direction.

**Auxiliary Information** Our RL-based acquisition policy builds on PPO (Schulman et al., 2017) and leverages the neural process model to provide auxiliary information and intermediate rewards. Table 4 illustrates the contributions of these components. The results clearly show that each component — both auxiliary information and the intermediate rewards — significantly enhances the acquisition policy, leading to better selection of informative prompts and more accurate benchmark performance estimation.

## 5 CONCLUSION

In this work, we present a novel approach for efficient LLM evaluation by leveraging dependency modeling and subset selection. Our key contributions include developing a generative model that captures dependencies across evaluation prompts and handles mixed-type evaluation scores, as well as proposing new subset selection policies based on these dependencies. Extensive experiments on multiple LLM evaluation benchmarks demonstrate the superiority of our RL-based acquisition policy in providing accurate benchmark performance estimation with a minimal acquisition budget. Our results also emphasize the importance of robustness to model bias and the effectiveness of our approach in cold start scenarios. Future research could explore integrating continual learning frameworks to enhance performance in the presence of model bias and cold starts. Additionally, expanding our methods to other benchmarks and refining the neural process model with improved uncertainty estimation are promising areas for further investigation.

---

[2]At the time of writing this paper, the average scores from the MTEB English leaderboard. (https://huggingface.co/spaces/mteb/leaderboard) for these four models are: SFR (67.56), E5 (66.63), BGE-large (64.23), and BGE-small (62.17).

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

## A NEURAL PROCESS

For a benchmark $X$ with $N$ prompts, $X = \{x_n\}_{n=1}^N$, we first use a pretrained embedding model to extract the representations for each prompt. During training, given a model $m$ with evaluation scores $Y_m$, we randomly select a subset of scores $Y_m^{(o)}$ as observed and maximize the log-likelihood for the remaining scores $Y_m^{(u)}$ based on the equation equation 3. When $Y_m^{(u)}$ is too large to fit into memory, we further sample a smaller subset from $Y_m^{(u)}$. Due to the inherent permutation invariance of the neural process model, random sampling will not affect the learning of dependencies across prompts.

### A.1 ARCHITECTURE

The neural process model consists of a prior network $p(z_m \mid Y_m^{(o)}, X)$, a posterior network $q(z_m \mid Y_m^{(u)}, Y_m^{(o)}, X)$, and a decoder $p(Y_m^{(u)} \mid z_m, Y_m^{(o)}, X)$. We generally follow the architecture of Attentive Neural Process (Kim et al., 2019), but replace the self-attention layer with a more memory-efficient Set Transformer layer (Lee et al., 2018). We also share the same network for both the prior and posterior. Before feeding the prompt embeddings into the prior/posterior network, we use an additional linear layer to reduce the dimensionality of the extracted representations. Similarly, the evaluation scores are passed through a linear layer to increase their dimensionality. We then concatenate the prompt representation with the score representation along the feature dimension and pass the concatenated set of vectors through a series of permutation equivariant Set Transformer layers. The outputs are then aggregated across the set elements to obtain a feature representation for the entire set. Following Set Transformer approach, we use learned pooling by multihead attention. The set representation is then passed through a linear linear to obtain the parameters for the latent distribution, which we assume to be Gaussian here. Please see Fig. 1(a) for an illustration of the prior/posterior network.

The decoder network $p(Y_m^{(u)} \mid z_m, Y_m^{(o)}, X)$ uses a cross-attention layer to produce a permutation equivariant representation for each prompt. In this layer, the query is the representation for $X$, the key is the representation for $X^{(o)}$, and the value is the permutation equivariant representation corresponding to $Y_m^{(o)}$ from the prior network. The permutation equivariant representation for each prompt is then concatenated with the prompt representation and the latent vector. These concatenated inputs are processed through a series of Set Transformer layers. The final outputs are then passed through a linear layer to predict the evaluation scores. Please see Fig. 1(b) for an illustration of the decoder network.

**Mixed-type Evaluation Scores** The above architecture uses a linear layer to obtain the representation for the evaluation scores. However, the linear layer is not suitable for discrete scores. Instead, we use an Embedding layer to represent the categorical evaluation scores. When a benchmark contains mixed-type scores, meaning some datasets report real-valued metrics while others report discrete scores, we additionally include an embedding vector to indicate the metric types.

### A.2 HYPERPARAMETERS

Table A.1 summarizes the hyperparameters used for the neural process model for each dataset. For the HELM-Lite and Chatbot Arena benchmarks, due to their relatively small number of models with evaluation scores, a neural process model with set transformer layers can easily overfit the data. Therefore, we use linear layers instead of the set transformer layers. Note that we did not conduct a thorough hyperparameter search. It is possible to further improve the results with optimized hyperparameters.

## B COMBINATORIAL OPTIMIZATION BASED ACQUISITION POLICY

Given the model $p(Y_m^{(u)} \mid Y_m^{(o)}, X)$, a static acquisition policy can be derived by searching over the training set to find the optimal subset of prompts that gives the most accurate prediction of the remaining prompts. This is a typical combinatorial optimization problem, which is NP-Hard. Here,

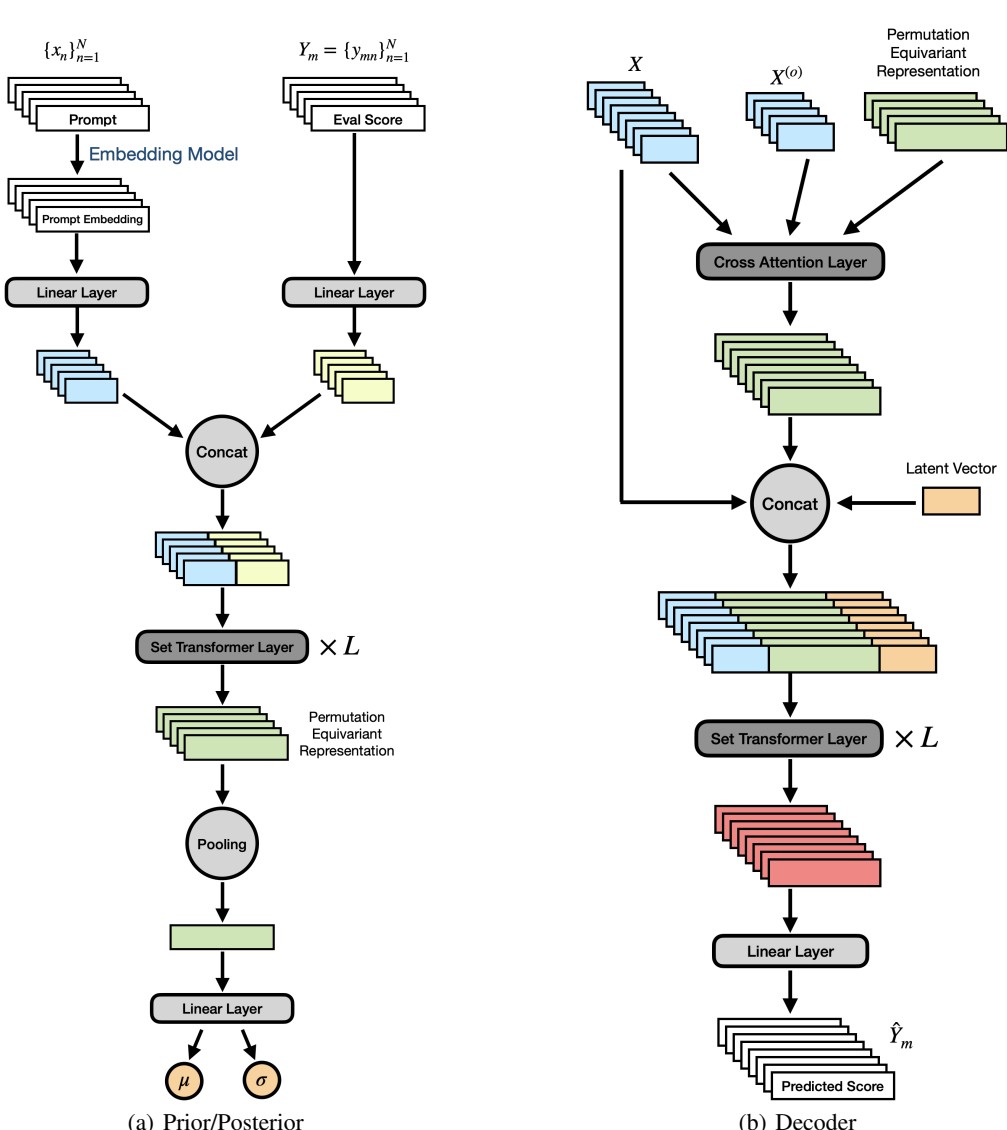

(a) Prior/Posterior                    (b) Decoder

Figure A.1: The architecture of the neural process model.

Table A.1: Hyperparameters for the nueral process model.

| | AlpacaEval | MMLU | Open LLM | HELM-Lite | Chatbot Arena |
|---|---|---|---|---|---|
| representation dimension for $x$ | 16 | 16 | 16 | 16 | 16 |
| representation dimension for $y$ | 16 | 16 | 16 | 16 | 16 |
| feature dimension for permutation equivariant layer | 32 | 32 | 32 | 32 | 16 |
| number of permutation equivariant layers for encoder | 1 | 2 | 2 | 1 | 1 |
| number of permutation equivirant layers for decoder | 1 | 2 | 2 | 1 | 1 |
| number of attention heads | 8 | 8 | 8 | N/A | N/A |
| number of induced points | 16 | 16 | 8 | N/A | N/A |
| latent dimension | 16 | 32 | 32 | 16 | 8 |

we employ a sequential approach that selects one prompt at a time until $K$ prompts are selected. Starting from an empty set $o = \emptyset$, the next prompt $i \in u := \{1, \dots, N\} \setminus o$ is chosen to minimize the prediction error over the training set, i.e.,

$$i = \arg\min_{i' \in u} \mathbb{E}_{Y_m \sim p_{\mathcal{D}}} \mathbb{E}_{\hat{Y}_m^{(u')} \sim p(Y_m^{(u')} | Y_m^{(o')}, X)} \| \hat{Y}_m^{(u')} - Y_m^{(u')} \|^2, \tag{B.1}$$

where $o' = o \cup \{i'\}$ and $u' = u \setminus \{i'\}$. We estimate the expectation by Monte Carlo sampling. For notation simplicity, the above equation computes the mean squared error on prompts $u'$; however, in practice, different datasets may use different metrics. Additionally, these differences may be weighted depending on the dataset size. Please refer to Algorithm 2 for pseudo-code of the selection process. Note that this approach has a complexity of $O(KMN)$, which could be prohibitive when the benchmark is large.

---

**Algorithm 2** Static Evaluation Acquisition via Combinatorial Optimization

**Require:** Acquisition budget $K$, Training set $\mathcal{D}_{train}$, Number of samples $S$, Neural Process $p$
1: $o = \emptyset, u = \{1, \dots, N\}$
2: **while** $|o| < K$ **do**
3:     $L = \{\}$
4:     **for** $i' \in u$ **do**
5:         $o' = o \cup \{i'\}, u' = u \setminus \{i'\}$
6:         Sample $S$ predictions $\{\hat{Y}_{m,s}^{(u')}\}_{s=1}^S$ from $p(Y_m^{(u')} | Y_m^{(o')}, X)$ for each model $m$
7:         $L[i'] = \frac{1}{|\mathcal{D}_{train}| \times S} \sum_{m=1}^{|\mathcal{D}_{train}|} \sum_{s=1}^S \| \hat{Y}_{m,s}^{(u')} - Y_m^{(u')} \|^2$
8:     **end for**
9:     $i = \arg\min_{i' \in u} L[i']$
10:     $o = o \cup \{i\}, u = u \setminus \{i\}$
11: **end while**

---

## C  REINFORCEMENT LEARNING BASED ACQUISITION POLICY

The acquisition policy determines the next prompt to acquire its evaluation score based on the current state, which includes the prompts $X^{(o)}$ and their scores $Y_m^{(o)}$ that have already been acquired. We further incorporate the candidate prompts $X^{(u)}$ into the policy inputs, i.e., $P(i \mid Y_m^{(o)}, X)$, so the policy has access to the action space. Including the candidate prompts in the inputs is crucial in the cold start setting since the action space differs between training and testing. Similar to the neural process model, the policy network employs two linear layers to obtain representations for both the prompts and the evaluation scores, which are then concatenated along the feature dimension. For the candidate prompts without available evaluation scores, we use a special embedding vector. Then, a permutation-invariant network processes the set of concatenated representations and outputs a aggregated representation for the entire set. We utilize the Set Transformer architecture for the permutation invariant network. Two branches of linear layers are added on top of the set representation for actor and critic, respectively. The actor branch outputs a vector with the same dimensionality as the prompt representations. The probability of selecting a prompt is proportional to the inner product of the output vector and the prompt representations. To prevent the policy from selecting duplicate prompts, the probability of the already selected prompts is manually set to zero. The critic branch outputs a scalar indicating the value estimation for the current state. Table C.1 summarizes the hyperparameters used for the policy network and PPO training process. We did

Table C.1: Hyperparameters for RL-based acquisition policy.

| | | |
|---|---|---|
| Policy Network | representation dimension for $x$ | 16 |
| | representation dimension for $y$ | 16 |
| | feature dimension for permutation equivariant layer | 32 |
| | number of permutation equivariant layers | 1 |
| | number of linear layers for actor | 1 |
| | number of linear layers for critic | 1 |
| PPO | advantage $\lambda$ | 0.95 |
| | discount factor $\gamma$ | 0.99 |
| | PPO clip range | [0.8, 1.2] |
| | entropy coefficient | 0.0 |

not conduct hyperparameter optimization and used the same set of hyperparameters for all datasets. Further improvements are likely possible with hyperparameter optimization tailored to each dataset.

## D  COLD START PROBLEM

In the cold start setting, the benchmark is expanded with new prompts for which no evaluation scores are initially available for any model. That is, the original benchmark $X = \{x_n\}_{n=1}^N$ have evaluation scores $Y_m = \{y_{mn}\}_{n=1}^N$ for $M$ models, while a set of new prompts $X' = \{x_n\}_{n=N+1}^{N'}$ do not have any evaluation scores.

To enable the neural process model generalize to the newly added prompts, we propose a semi-supervised training procedure, where the new prompts are treated as unlabeled data. During training, we optimize the log-likelihood equation 3 for $X$ and $Y_m$. Simultaneously, we predict the evaluation scores for the new prompts $X'$ based on the current trained model. When the prediction is sufficiently accurate, meaning the uncertainty is lower than a predefined threshold, we add the predicted scores as synthetic training data to optimize the ELBO equation 3.

The RL policy in the cold start setting follows a similar architecture to Sec. C. To help the policy generalize to unseen prompts, we use the learned prompt representations from the neural process model and keep them fixed throughout the training process. Additionally, we found that entropy regularization over the actor distribution aids generalization, which is set to 0.001 in our experiments.

## E  EXPERIMENTS

### E.1  LLM LEADERBOARD

We conduct experiments on 5 popular LLM benchmarks:

- **HuggingFace Open LLM Leaderboard** (Beeching et al., 2023) consists of 6 datasets with a total of 28,659 prompts. Evaluation scores include both binary accuracy and real-valued probabilities. We collect evaluation scores for 2,084 models and select 1,000 models for training based on their evaluation date. The most recently evaluated models are used for testing, simulating the real-world scenario.

- **MMLU** (Hendrycks et al., 2020) contains 57 datasets with a total of 14,042 multiple choice QA problems on different subjects. Evaluation scores are all binary accuracy. We collect evaluation scores for the same models from the Open LLM Leaderboard.

- **HELM-Lite** (Liang et al., 2022) include 10 datasets (each possibly containing several sub-datasets) with a total of 13,021 prompts. Evaluation scores include both binary exact match scores and real-values metrics such as F1 and BLEU. We collect evaluation scores for 33 models and randomly select 23 models for training since the evaluation does not have dates.

- **AlpacaEval 2.0** (Li et al., 2023) contains 805 prompts. For each model, the generations are compared to those of GPT-4 to compute the win rate. Although this benchmark is relatively small,

Table E.1: Benchmark performance estimation error on each LLM benchmark. Lower is better.

| | AlpacaEval (K=100) | HELM-Lite (K=200) | Open LLM (K=200) | MMLU (K=100) | Chatbot Arena (K=40) |
|---|---|---|---|---|---|
| Uniform | $0.005 \pm 0.000$ | $0.038 \pm 0.005$ | $0.022 \pm 0.002$ | $0.018 \pm 0.001$ | $0.052 \pm 0.010$ |
| S-Rand | - | $0.035 \pm 0.012$ | $0.030 \pm 0.003$ | $0.017 \pm 0.002$ | - |
| C-Embed | $0.006 \pm 0.001$ | $0.051 \pm 0.010$ | $0.024 \pm 0.002$ | $0.020 \pm 0.004$ | $0.052 \pm 0.014$ |
| C-Score | $0.004 \pm 0.001$ | $0.054 \pm 0.010$ | $0.031 \pm 0.003$ | $0.014 \pm 0.002$ | $0.054 \pm 0.004$ |
| C-IRT | $0.003 \pm 0.001$ | $0.044 \pm 0.013$ | $0.026 \pm 0.001$ | $0.015 \pm 0.001$ | $0.057 \pm 0.002$ |
| Comb-Optim | $0.006 \pm 0.003$ | - | - | - | $0.065 \pm 0.012$ |
| Uncertainty | $0.011 \pm 0.001$ | $0.055 \pm 0.013$ | $0.063 \pm 0.015$ | $0.050 \pm 0.003$ | $0.035 \pm 0.003$ |
| LatentInfoGain | $0.010 \pm 0.003$ | - | - | - | $0.066 \pm 0.025$ |
| RL | $\mathbf{0.001 \pm 0.000}$ | $\mathbf{0.030 \pm 0.005}$ | $\mathbf{0.018 \pm 0.001}$ | $\mathbf{0.013 \pm 0.000}$ | $\mathbf{0.034 \pm 0.006}$ |

it requires an expensive GPT-4 based judge, so reducing the number of API calls can significantly reduce the total evaluation cost. We collect evaluation scores for 130 models and randomly select 70% for training.

- **Chatbot Arena** (Zheng et al., 2024) is a popular human-annotated benchmark, where annotators interact with two anonymous models using the same prompts and declare a winner. We use the pairwise comparisons evaluated on the 80 MTBench prompts (Zheng et al., 2024). Although this benchmark is relatively small, human evaluation is expensive, so further reducing the evaluation prompts could lower costs. The annotations include comparisons over multiple turns, but we only use the annotations for the first turn here. Unlike other benchmarks where each model directly receives an evaluation score, this benchmark evaluates each pair from a set of 6 models. To create the train-test splits, we randomly select one of the six models, and all pairs that involve the selected model are included in the test split. Note that not all 80 prompts are annotated for each model pair. While our neural process model can handle missing data, the acquisition process must acquire the true score for any prompt the policy selects. To address the missing data during acquisition process, we use the trained neural process to predict the missing evaluation scores. We report win rate for this benchmark.

## E.2 EVALUATION PROCEDURE

For a model $m'$ to be evaluated, the acquisition policy determines a subset of prompts $X^{(o)}$ to acquire the true evaluation scores. The neural process model $p(Y_{m'}^{(u)} \mid Y_{m'}^{(o)}, X)$ predicts the evaluation scores for the remaining prompts. The benchmark performance is then estimated based on these predicted scores. For benchmarks with only one dataset, the benchmark performance is the average over all examples. For benchmarks with multiple datasets, the benchmark performance is averaged over the performance of each dataset. For example, the HuggingFace Open LLM Leaderboard consists of 6 datasets, so the benchmark performance is the average of the performance on these 6 datasets. The MMLU dataset further contains 57 subsets, so its performance is the average over these 57 subsets. For Chatbot Arena, we report win rate as the benchmark performance. For final evaluation results, we compute the absolute difference between the predicted benchmark performance and the real benchmark performance for each model in the test split and report the average absolute error over all models in the test split.

## E.3 ADDITIONAL RESULTS

Table E.1 presents the benchmark performance estimation errors for various acquisition policies across different LLM benchmarks. We conduct experiments with 3 random seeds for each benchmark and report the average estimation error and standard deviation under the specified acquisition budget. Prompt embeddings are obtained using the SFR embedding model. For the AlpacaEval and Chatbot Arena benchmarks, stratified random sampling is equivalent to uniform sampling since they only contain one dataset. Combinatorial optimization and information gain based policies are too expensive to run for HELM-Lite, HuggingFace Open LLM Leaderboard, and MMLU due to the large number of prompts in each benchmark.

The RL-based acquisition policy consistently achieves the lowest error across all benchmarks, indicating its superior ability to select informative prompts and accurately estimate benchmark performance. The stratified random sampling performs similarly to the uniform sampling, and these

random acquisition policies generally are competitive, particularly because they are efficient and do not rely on any other models to determine the prompt selection.

The Cluster-Embed policy does not perform any better than the random selection, suggesting that the similarity in prompt embedding does not always correlate with the similarity in the evaluation scores. Utilizing evaluation scores for clustering shows mixed results. The Clustering-Score policy outperforms Clustering-Embed on AlpacaEval and MMLU but underperforms on HELM-Lite, Open LLM and Chatbot Arena benchmarks. Clustering based on IRT features generally provides better performance estimation since these features are learned to reflect the evaluation scores.

The combinatorial optimization based policy does not perform well, even on the two small benchmarks where it is computationally feasible. We attribute this to a potential distribution shift between the models used for training and those used for testing, suggesting that the static policy optimized on training models does not generalize well to new models during testing.

The uncertainty sampling based acquisition policy does not perform well across all benchmarks. Theoretically, the uncertainty sampling method requires a good estimation of the aleatoric uncertainty to perform well. However, in practice, the uncertainty from the neural process model combines the aleatoric and epistemic uncertainties. Quantifying and decomposing the aleatoric and epistemic uncertainties is an active research ares in machine learning (Gawlikowski et al., 2023; Wimmer et al., 2023; Hüllermeier & Waegeman, 2021), which we leave for future work to explore for our AEA application. Similarly, the information gain based acquisition policy also requires accurate uncertainty estimation, which is challenging, especially with scarce training data on AlpacaEval and Chatbot Arena benchmarks.