# OpenReview forum: "Active Evaluation Acquisition for Efficient LLM Benchmarking"
_ICLR.cc/2025/Conference — Submitted to ICLR 2025_

### Official Review · Reviewer_zhCw · 2024-10-29

**Soundness:** 2
**Presentation:** 3
**Contribution:** 2
**Rating:** 6
**Confidence:** 3

**Summary:**

Active Evaluation Acquisition (AEA) is a method that selects a subset of samples from various benchmarks to efficiently evaluate a model's performance on the benchmarks. They test various confidence- and density-based metrics for selecting subsets of benchmark data, as well as create an RL-based dynamic selection policy. Given the current selected subset and a data point $i$, the policy is used to estimate the performance of the model on $i$, which is then sampled multiple times to determine the confidence of the estimation. They test their method on popular datasets (including AlpacaEval, MMLU, and Chatbot Arena) with popular models (including GPT-4, LLaMa, and Mistral).

**Strengths:**

The paper is well written, and well structured. Their selected baselines are appropriate and quite thorough, including metrics from information theory to classic clustering-based metrics. They also show great thought into the methodology by addressing cold start. The appendix also outlines helpful information, such as Algorithm 2. Finally, the small detail of including standard deviations for each of their results showcases the stability of their algorithm.

**Weaknesses:**

This is an interesting idea and innovative approach - I am having some trouble understanding why choosing samples with less confidence in evaluation will result in a rigorous enough evaluation. The baselines have a clear objective: choose representative samples. Your paper can be stronger if you (1) clarify the objective of your RL-based method, and (2) include more analysis on why RL achieves near 0% absolute error on AlpacaEval. Also, only AlpacaEval achieves such results - the other datasets have an error of ~2%. Including a comparative analysis on the performance of AppacaEval versus the other datasets can help readers understand the discrepancy in performance. Finally (this is more of a nitpicky thing), consider calling your RL-based method RL-AEA to make clear your contribution. Or at least append "RL" with "(ours)" in Figures 1-2 and Tables 2-3, E.1.

**Questions:**

I have two questions:
1. Are there any recent "Active Testing" works with language models? Kossen et al is from 2021, and is tested on Fashion-MNIST and CIFAR-100.
2. Could you motivate the use of an RL agent to estimate the value of a data point? Why are you using an RL agent to calculate the model uncertainty? While the formulation makes sense, a few more baselines would strengthen the motivation. Baselines can include: (1) self-consistency (sample the model multiple times for a response, if the response variety is large, classify the data sample as "not confident") or (2) model feedback (use the distribution over the tokens or perplexity to determine confidence).

---

> ### Author Response · Authors · 2024-11-14
>
> Thank you for the review and insightful feedback. We appreciate the recognition of the well-written and structured nature of our paper, as well as the appropriate selection of baselines and the effort put into addressing the cold start problem. We will address the weaknesses and questions raised in this rebuttal.
>
> **Let us clarify the core problem and approach upfront:**
>
> The primary goal of our work is to improve the efficiency and reduce the substantial computational cost associated with evaluating LLMs on comprehensive benchmarks comprising thousands or millions of prompts.
>
> Our approach tackles this challenge through two key components: 1) Subset selection, where we actively select the most informative subset of prompts for evaluation, and 2) Performance prediction, where we leverage dependencies across prompts to predict the LLM's performance on the remaining, unselected prompts without running actual inference on those prompts.
>
> To give a concrete example, let's say a benchmark has $N$ prompts. The subset selection policies (including random, clustering-based, uncertainty-based, and RL-based policies) will select $K$ prompts based on their corresponding strategies. Then we will run LLM inference on these $K$ prompts to acquire the actual evaluation scores. Afterwards, the evaluation scores on the remaining $N-K$ prompts will be predicted by the neural process model.
>
> By combining these two components, our method aims to provide reliable estimates of an LLM's overall benchmark performance while minimizing the number of prompts that require expensive LLM inference. This enables significant computational savings and reduces the associated costs.
>
> **Here are detailed responses to the questions you raised:**
>
>
> > I am having some trouble understanding why choosing samples with less confidence in evaluation will result in a rigorous enough evaluation
>
> The high uncertainty of the prediction model means the performance prediction on these prompts is not accurate. By selecting these prompts to acquire their actual evaluation scores, the overall estimation error will be lower. This is the same principle as in active learning literature, where the uncertain examples based on the current model are selected to further train the model.
>
> > clarify the objective of your RL-based method
>
> The objective of the RL-based policy is the same as other selection policies, that is, to select informative examples. The RL agent does so by maximizing the accumulated reward, which is defined as how accurately we can recover the benchmark performance using the selected prompts (see line 242). We will clarify the RL objective in the camera-ready version.

---

> ### Author Response · Authors · 2024-11-14
>
> > include more analysis on why RL achieves near 0% absolute error on AlpacaEval. Also, only AlpacaEval achieves such results - the other datasets have an error of ~2%. Including a comparative analysis on the performance of AlpacaEval versus the other datasets can help readers understand the discrepancy in performance.
>
> One potential reason for the exceptional performance on AlpacaEval could be the relatively smaller size of the benchmark compared to others. With fewer prompts, the dependencies across prompts may be easier to capture, leading to more accurate predictions.
>
> Additionally, the scores on AlpacaEval 2.0 are computed by a logistic regression model that predicts the win rate from a GPT4 evaluator. These scores are essentially smoother than the typical accuracy scores in other leaderboards.
>
> We will conduct a detailed analysis to investigate the factors that contribute to the varying performance across different benchmarks, such as the benchmark size, prompt diversity, and the complexity of the underlying tasks. This analysis will provide valuable insights and help us better understand the strengths and limitations of our approach in different evaluation contexts.
>
> > Could you motivate the use of an RL agent to estimate the value of a data point? Why are you using an RL agent to calculate the model uncertainty? While the formulation makes sense, a few more baselines would strengthen the motivation. Baselines can include: (1) self-consistency (sample the model multiple times for a response, if the response variety is large, classify the data sample as "not confident") or (2) model feedback (use the distribution over the tokens or perplexity to determine confidence).
>
> The RL agent is used to select K prompts out of the total N prompts to acquire their actual evaluation scores. The evaluation scores on the remaining N-K prompts will be predicted by the neural process model. For the RL-based policy, uncertainty on the unselected prompts is implicitly handled by the policy. That is, the RL policy implicitly estimate the uncertainty of the available prompts and choose the one that maximize the long-term accumulated reward. This is in contrast to a myopic greedy policy that always select the most uncertain one.
>
> Thank you again for the review and feedback. We hope that with the clarifications and improvements outlined above, you would consider increasing the rating for our paper.

---

> ### Comment · Reviewer_zhCw · 2024-11-19
>
> Thank you for the clarifications in the first comment - all those are resolved on my end.
>
> Regarding your second comment:
>
> 1. "AlpacaEval could be the relatively smaller size", "scores on AlpacaEval 2.0 are computed by a logistic regression" - understood, resolved.
>
> 2. "We will conduct a detailed analysis to investigate" - could you please show some analysis for the review? It can be preliminary, but it will be hard to be convinced without *some* analysis.
>
> 3. For my question "Why are you using an RL agent to calculate the model uncertainty?", I moreso meant why are you using an RL agent versus other common techniques to measure uncertainty? With your paper and above clarifications, I understand that using an RL agent saves inference computation and is cheaper. My main doubt is: *does it perform better than inference-dependent methods*? Hence, I suggested two baselines (self-consistency, and model feedback) that can help solidify your motivation for using an RL agent.
>
> 4. If you could also clarify whether there are any recent "Active Testing" works with language models (Question 1 in my review), that would be helpful too.

---

> > ### Author Response · Authors · 2024-11-24
> >
> > **Related active testing works**
> >
> > Regarding your question about recent "Active Testing" works with language models, we have conducted a thorough literature review and found a few relevant works:
> >
> >
> > 1. "Active Testing of Large Language Model via Multi-Stage Sampling" (Huang et al., 2024) propose AcTracer, a multi-stage sampling technique. Given unlabeled test examples, AcTracer clusters them based on LLM internal embeddings, selects a cluster based on its variance estimation, and then selects a test example within that cluster based on the LLM's generation confidence. The sampled examples are labeled to estimate the LLM's performance.
> > 2. “Label-Efficient Model Selection for Text Generation” (Ashury-Tahan et al., 2024) propose DiffUse to select preference data to obtain their oracle preference labels. They first obtain the embeddings of each model and then take their differences, then the embedding differences are clustered and one representative data is selected from each cluster.
> > 3. “Efficiently Measuring the Cognitive Ability of LLMs: An Adaptive Testing Perspective” (Zhuang et al., 2024) propose an adaptive testing framework that dynamically adjusts the difficulty of the test questions based on the LLM’s performance, enabling more accurate estimation of the model’s abilities using fewer questions. While conceptually similar to our AEA approach, their approach produces a scalar ability score for each model, which is hard to interpret. On the contrary, our AEA produces the comprehensive evaluation scores for all prompts as for traditional benchmarks.
> >
> >
> > Beyond “Active Testing” area, there are several other works exploiting the “active selection” ideas:
> >
> >
> > 1. “Active Prompting with Chain-of-Thought for Large Language Models” (Diao et al., 2024) propose to actively select in context examples with CoT reasoning to adapt LLMs to different tasks.
> > 2. “Deep Bayesian Active Learning for Preference Modeling in Large Language Models” (Melo et al. 2024) utilize bayesian active learning principles to select preference training data. They advocate for selecting data points with high epistemic uncertainty according to the preference model and simultaneously maximizing the entropy of selected prompts in the feature apace.
> > 3. “LESS: Selecting Influential Data for Targeted Instruction Tuning” utilizes gradient information to select instruction tuning data that can help a target task.
> >
> > We will incorporate these relevant works into our related work section and discuss how our AEA approach differs in objective and methodology.

---

> > > ### Comment · Reviewer_zhCw · 2024-11-24
> > >
> > > I appreciate the extent of details in these additions. These were great experiments, and all my concerns are addressed - I've raised my score. Good luck!

---

> ### Author Response · Authors · 2024-11-24
>
> Thank you for the additional comments and questions. We appreciate the opportunity to further clarify our work and address your remaining concerns.
>
> **Preliminary analysis on performance prediction accuracy**
>
> To investigate potential factors that can impact the accuracy of predicting evaluation scores, we conducted a preliminary analysis on the HELM-Lite benchmark due to its diverse set of datasets covering different types of tasks and metrics. To eliminate any potential impact from the prompt selection policy, this analysis was performed using randomly selected 50 prompts as condition to predict the evaluation scores.
>
> It's worth noting that prompts that are inherently difficult for language models, resulting in consistently low scores, do not necessarily translate to high prediction error for the evaluation scores. For example, if no model can solve a particular prompt, its evaluation score will always be zero, making it relatively easy to predict.
>
> For a dataset with $N$ prompt evaluated on $M$ models from training set, we denote the evaluation scores as $S \in \mathbb{R}^{N \times M}$. We consider the following factors:
>
> * Metric Types: Our hypothesis is that discrete metrics, such as exact match accuracy, are harder to predict compared to smooth continuous metrics like BLEU.
> * Prompt Diversity: A dataset with diverse prompts is potentially harder for LLM. We estimate the diversity using the pairwise similarity of the prompt embeddings.
> * Task Difficulty: We estimate the difficulty of a task as 1 minus the average evaluation scores across all models, i.e., $ (1-S).mean()$. A higher value indicates a more difficult task on average.
> * Score Diversity: A dataset where the score distribution has a high variance can potentially lead to higher prediction error. We calculate score diversity as $S.var(dim=0).mean()$
> * Task Informativeness: We estimate the informativeness of each prompt as the variance of its evaluation scores on all models, then the task informativeness is averaged over prompts. $S.var(dim=1).mean()$.
> * Evaluation Variability: Calculating the variance of the mean scores across prompts is another way to quantify the variability or diversity in the task. $S.mean(dim=0).var()$
>
> Please see Table R.1 for detailed results on 28 subsets from HELM-Lite. In Table R.2, we calculate the correlation between each factor and the prediction error.
>
> *Table R.2: Correlation between dataset characteristics and prediction error.*
>
> |    | factor               |   spearman |   pearson |
> |---:|:---------------------|-----------:|----------:|
> |  0 | metric_type          |   0.78296  |  0.759027 |
> |  1 | prompt_diversity     |  -0.722215 | -0.718767 |
> |  2 | score_diversity      |   0.215654 |  0.366019 |
> |  3 | task_informativeness |   0.719759 |  0.780538 |
> |  4 | eval_variability     |   0.88451  |  0.841635 |
> |  5 | task_difficulity     |  -0.195402 | -0.138457 |
>
>
> We can observe that:
>
> 1. Metric Type has a high correlation with prediction error, verifying our hypothesis. We further analyzed the differences across metric types in Table R.3 and Table R.4.
> 2. Prompt Diversity has a high negative correlation with the prediction error. Although counterintuitive at first, this aligns with the analysis that harder task for LLM does not necessarily mean harder evaluation score prediction. This is further supported by the negative correlation between Task Difficulty and prediction error.
> 3. Score Diversity only has a moderate correlation to prediction error.
> 4. Task Informativeness has a high correlation with prediction error, as expected, since prompts with high variance in evaluation scores are inherently harder to predict accurately.
> 5. Evaluation Variability is actually an estimation of the prediction error on test set using the scores from training set, so  its high correlation with prediction error is unsurprising.

---

> ### Author Response · Authors · 2024-11-24
>
> To further investigate the impact of metric types, we analyzed the prediction errors for different metric types in Tables R.3 and R.4. The prediction errors for binary evaluation scores are significantly higher than those for real-valued scores like BLEU_4 and F1_score, aligning with our hypothesis.
>
> *Table R.3: Prediction error for different metric types.*
>
> | metric_type   |   prediction_error |
> |:--------------|-------------------:|
> | binary        |          0.105535  |
> | real          |          0.0333905 |
>
> *Table R.4: Prediction error for different metrics.*
>
> | metric_name                 |   prediction_error |
> |:----------------------------|-------------------:|
> | bleu_4                      |          0.0302389 |
> | exact_match                 |          0.0776866 |
> | f1_score                    |          0.0386433 |
> | final_number_exact_match    |          0.113437  |
> | math_equiv_chain_of_thought |          0.136722  |
> | quasi_exact_match           |          0.0956804 |
>
>
> This preliminary analysis provides valuable insights into the factors that can influence the accuracy of predicting evaluation scores. We plan to conduct a more comprehensive investigation across multiple benchmarks and develop strategies to improve prediction accuracy, especially for challenging metric types and datasets with high variability or informativeness.

---

> ### Author Response · Authors · 2024-11-24
>
> **Uncertainty estimation baselines**
>
> Regarding your question about comparison with typical LLM uncertainty estimation approaches. You raise a good point that comparing with these inference-dependent uncertainties can further strengthen our paper.
>
> As a preliminary comparison, we compared with two baselines: 1) directly use the generation log-likelihood as uncertainty estimate, and 2) semantic entropy (Kuhn et al., 2023), where multiple generations are clustered based on a pretrained NLI model, and afterwards, the entropy is calculated to incorporate the semantic meanings of the clusters.
>
> Table R.4 presents the results on three models for estimating their performance on AlpacaEval benchmark. For each model, we use either our RL policy or these uncertainty estimations to acquire the evaluation scores for a subset of prompts. The evaluation scores on the remaining prompts are estimated by the same neural process model. From the results, we can see that our RL-based acquisition policy consistently performs better than these two uncertainty estimates.
>
> *Table R.4: Comparison of our RL-based acquisition policy and uncertainty-based sampling (using generation perplexity and semantic entropy) in estimating AlpacaEval benchmark performance. We compare the absolute error with selection budget of 10, 20, 50 and 100.*
>
> |                                                        |     10 |     20 |     50 |    100 |
> |:-------------------------------------------------------|-------:|-------:|-------:|-------:|
> | ('Mistral-7B-Instruct-v0.2', 'Perplexity')             | 0.0162 | 0.0156 | 0.0103 | 0.0088 |
> | ('Mistral-7B-Instruct-v0.2', 'Semantic Uncertainty')   | 0.0122 | 0.0175 | 0.0126 | 0.015  |
> | ('Mistral-7B-Instruct-v0.2', 'RL (ours)')              | 0.0012 | 0.0008 | 0.0006 | 0.0006 |
> | ('Mixtral-8x7B-Instruct-v0.1', 'Perplexity')           | 0.0025 | 0.0053 | 0.0058 | 0.0076 |
> | ('Mixtral-8x7B-Instruct-v0.1', 'Semantic Uncertainty') | 0.0095 | 0.0145 | 0.0132 | 0.0202 |
> | ('Mixtral-8x7B-Instruct-v0.1', 'RL (ours)')            | 0.0019 | 0.0003 | 0.0003 | 0.0002 |
> | ('gemma-7b-it', 'Perplexity')                          | 0.009  | 0.019  | 0.0136 | 0.0147 |
> | ('gemma-7b-it', 'Semantic Uncertainty')                | 0.0148 | 0.0139 | 0.0144 | 0.0169 |
> | ('gemma-7b-it', 'RL (ours)')                           | 0.0013 | 0.0007 | 0.0009 | 0.0004 |
>
>
> Our hypothesis for why the RL approach outperforms these inference-dependent methods is that the RL policy is trained jointly with the neural process model to align its uncertainty estimates. This allows it to acquire prompts where the neural process predictions are most uncertain or inaccurate. In contrast, while the inference-dependent uncertainties may capture the model's own uncertainty well, they can be misaligned with the uncertainties of the neural process model used for prediction, resulting in overall higher benchmark estimation errors.
>
> Furthermore, we want to emphasize that these inference-dependent uncertainty methods are not practical to deploy in reality, as they require running inference on all prompts and sometimes generating multiple responses to estimate uncertainty. This essentially increases the overall evaluation cost, defeating the purpose of efficient benchmarking.
>
> We will add these comparison in the appendix for the camera-ready revision.

---

### Official Review · Reviewer_P8Cj · 2024-11-04

**Soundness:** 3
**Presentation:** 3
**Contribution:** 3
**Rating:** 8
**Confidence:** 4

**Summary:**

The paper proposes investigating strategies for the efficient evaluation of LLMs. Their approach models dependencies across test examples using Neural stochastic processes and then uses a novel RL-based sampling policy that leverages the captured dependencies between test examples.

**Strengths:**

1) The Neural process model is novel and interesting in this context;
2) The application of RL is also novel in the context of efficient evaluation of LLMs. The idea seems promising;
3) The authors are able to get strong empirical results.

**Weaknesses:**

1) Section 2.2 can be greatly improved; it is not even clear how fitting is possible without observing labels Y^u. The authors cite VAE, but in VAEs the outputs are known, which makes the decoder training possible. This suggests that some assumptions about the model are not included (or clear) in the text. A more detailed explanation in the text would be appreciated (possibly including an algorithm box for model training);
2) I am not sure including Chatbot Arena makes sense here because it is a pairwise comparison dataset and $Y_m$ will depend on other models you're comparing against. Your NP model does not seem to accommodate that.
3) The paper does not comment on IRT adaptive testing on related work (see https://arxiv.org/pdf/2404.00712, for example). This class of methods is directly related to the paper and could be considered a competitor to the RL approach. It is important to mention why you do not compare against it;
4) The paper does not comment on limitations. One major limitation is the need to fit a neural net for every LLM we want to make inferences about. Moreover, as noted by tinyBenchmarks, adaptive testing can reduce the scope of applications as it can make efficient evaluation more costly;

Minor:
1) typo "Comparson";

**Questions:**

1) Does your model have any components trained using a training set? From the experiments section, it seems so, but this is not clear from the methods section.
2) In your equation 3 RHS, shouldn't we take the expectation wrt the unobserved portion of $Y$ (given that we do not observe it)?
3) Also from eq 3, it is not clear what are the assumptions on p(Y^u|...) and how it can be modeled because without observing Y^u;
4) As far as I understand, you need to refit your Neural Process model (or at least the posterior of Z_m) for every LLM $m$ you want to evaluate. How expensive is this step? One of the main applications of efficient evaluation of LLMs is quick evaluation during pre-training and it seems that your approach would not be friendly in this case;
5) Why do you compare against the Cluster-IRT approach and not IRT++ (from tinyBenchmarks)? IRT++ seems a better method;
6) It seems that some curves are missing in Fig 1, for example. Why? If they are not included on purpose, the legend should not include them.

---

> ### Author Response · Authors · 2024-11-14
>
> Thank you for the review and insightful feedback. We appreciate the recognition of the novelty and potential of our approach, as well as the strong empirical results. We will address the weaknesses and questions raised in this rebuttal.
>
> **Let us clarify the approach and training procedure upfront:**
>
> Our approach consists of two key components: 1) Subset selection policy, where we actively select the most informative subset of prompts for evaluation, and 2) Performance prediction, where we leverage dependencies across prompts to predict the LLM's performance on the remaining, unselected prompts without running actual LLM inference on those prompts.
>
> The training procedure first divide the leaderboard evaluation scores into train and test splits. The train split contains a set of models and their corresponding evaluation scores on the benchmark prompts. Specifically, for each model m in the train split, we have access to the full set of evaluation scores Y_m = \{y_{mn}\}, where n indexes over all N prompts in the benchmark. The test split contains another set of (non-overlapping) models and evaluation scores.
>
> During training, we utilize this train split to learn the neural process model and the subset selection policies:
>
> For the neural process model:
>
> * For each model m, we randomly sample two non-overlapping subsets from Y_m, treating one as the observed set Y_m^o and the other as the unobserved set Y_m^u.
> * The neural process is trained to optimize the ELBO objective, learning to predict the unobserved scores Y_m^u based on the observed scores Y_m^o, i.e. p(Y_m^u \mid Y_m^o, X). The ELBO is feasible to compute since train split have all scores available.
> * After training, the neural process model can predict the unobserved scores given an arbitrary subset of observed scores. It can generalize even to unseen models. Therefore, it does not require retraining on the test split.
>
>
> For the subset selection policies:
>
> * The random policy does not have trainable parameters, as prompts are selected at random
> * The Clustering-Embed policy does not have trainable parameters either. We simply use the prompt embedding to group the prompts.
> * The Clustering-Score policy directly use the evaluation scores for a prompt from the train split as its feature representation.
> * The Clustering-IRT policy trains an IRT model on train split, then we cluster the prompts based on the learned prompt embeddings from IRT models.
> * The Combinatorial Optimization policy, Uncertainty Sampling policy and Information Gain policy do not have trainable parameters either. They rely on the neural process model to predict the evaluation scores and their uncertainties.
> * The RL-based policy trains a selection policy on train split by interactively select which prompt to acquire its evaluation score.
>
> During testing, the subset selection policy selects a subset of prompts, whose actual evaluation scores will be acquired. Therefore, the actual evaluation scores are used for those prompts. While the evaluation scores for other prompts will be predicted by the neural process model. Since we have the gold evaluation scores for all prompts, we can compute the metrics by comparing the predicted scores to the gold scores.
>
> When the system is actually deployed, for a new model to be evaluated, the subset selection policy will select a subset of prompts, and we run the LLM inference on these prompts to get the real evaluation scores. The evaluation scores on the remaining prompts will be predicted by the neural process model.

---

> ### Author Response · Authors · 2024-11-14
>
> **Regarding your specific points:**
>
> > Section 2.2 can be greatly improved; it is not even clear how fitting is possible without observing labels $Y^u$. The authors cite VAE, but in VAEs the outputs are known, which makes the decoder training possible. This suggests that some assumptions about the model are not included (or clear) in the text. A more detailed explanation in the text would be appreciated (possibly including an algorithm box for model training);
>
>
> During training, the evaluation scores $Y_m^u$ are available for the models in train split. We will include a training algorithm, as described above, for the camera ready version to clarify these confusions.
>
>
> > I am not sure including Chatbot Arena makes sense here because it is a pairwise comparison dataset and will depend on other models you're comparing against. Your NP model does not seem to accommodate that.
>
>
> It’s conceptually similar to evaluation scores for a single model. Instead, now we have evaluation scores for a pair of model. For Chatbot Arena, the train split contains a set of model pairs and the win rate scores on each prompt.
>
>
> > The paper does not comment on IRT adaptive testing on related work (see https://arxiv.org/pdf/2404.00712, for example). This class of methods is directly related to the paper and could be considered a competitor to the RL approach. It is important to mention why you do not compare against it;
>
>
> You're absolutely right. We will include a discussion of IRT adaptive testing methods in the related work section and provide a comparison to our RL-based approach, highlighting the potential advantages and limitations of both approaches.
>
>
> > The paper does not comment on limitations. One major limitation is the need to fit a neural net for every LLM we want to make inferences about. Moreover, as noted by tinyBenchmarks, adaptive testing can reduce the scope of applications as it can make efficient evaluation more costly;
>
>
> We appreciate you highlighting these potential limitations. We will include a dedicated section in the paper to discuss them, including the potential increase in evaluation cost for certain applications, as noted by tinyBenchmarks. We will also discuss the trade-offs and scenarios where the overhead of our approach could outweigh the benefits.
>
> However, we want to emphasize that the bottleneck for LLM evaluation is on LLM inference, as it typically requires high-end hardware or calls to expensive APIs. Compared to the LLM itself, our neural process model and the subset selection policy are much smaller in size and computation (typically 2-3 linear layers in our experiments). Therefore, the overhead introduced by our method is relatively small compared to the computational savings achieved by reducing the number of prompts that require LLM inference.
>
> While our approach does involve additional components and training steps, the primary computational benefit stems from the significant reduction in the number of prompts that require running the resource-intensive LLM inference. In our experiments, we observed that our method could achieve accurate performance estimation while acquiring evaluation scores for only a small subset (e.g., <1% for MMLU and OpenLLM, 1.5% for HELM-Lite) of the total prompts.
>
>
> > Does your model have any components trained using a training set? From the experiments section, it seems so, but this is not clear from the methods section.
>
>
> Yes, our method does have components trained on a training set. As described above, the neural process model and the RL-based acquisition policy are trained on the train split of the leaderboard data.
>
>
> > In your equation 3 RHS, shouldn't we take the expectation wrt $Y^u$ the unobserved portion of (given that we do not observe it)?
>
> > Also from eq 3, it is not clear what are the assumptions on $p(Y^u|...)$ and how it can be modeled because without observing $Y^u$;
>
>
> The evaluation scores $Y^u$ are assumed available during training for the models in the train split. They will be predicted during testing for the models in the test split. We will clarify the assumptions and provide more detailed explanations in the revised manuscript.

---

> ### Author Response · Authors · 2024-11-14
>
> > As far as I understand, you need to refit your Neural Process model (or at least the posterior of $Z_m$) for every LLM m you want to evaluate. How expensive is this step? One of the main applications of efficient evaluation of LLMs is quick evaluation during pre-training and it seems that your approach would not be friendly in this case;
>
>
> We do not need to refit the neural process model for a new test model. As long as we acquire a subset of evaluation scores $Y^o$ following the subset selection policy, the neural process can predict the unobserved scores following $p(Y^u | Y^o, X)$.
>
>
>
> > Why do you compare against the Cluster-IRT approach and not IRT++ (from tinyBenchmarks)? IRT++ seems a better method;
>
>
> For notation clarity, we used the name Clustering-IRT, but we indeed used the strongest model recommend by tinyBenchmarks, which is IRT++. We will clarify this in the camera ready version.
>
>
> > It seems that some curves are missing in Fig 1, for example. Why? If they are not included on purpose, the legend should not include them.
>
>
> The legend is shared by all 5 plots to avoid cluttering, but for some benchmarks, such as HELM-Lite and OpenLLM, where the number of total prompts is large, Combinatorial Optimization and Information Gain policies are too expensive to finish in a reasonable time frame. Therefore, we left them out for those benchmarks. We will clarify this in the figure caption or legend.
>
>
> We hope that by addressing these weaknesses and questions, we have strengthened the clarity, rigor, and positioning of our work. Please let us know if you have any further comments or suggestions.

---

> ### Comment · Reviewer_P8Cj · 2024-11-19
>
> Thank you for all of your comments. You have a good paper but that needs more polishing, especially in the writing. I will increase my score, but please implement all the promised changes before the paper is published.

---

> > ### Author Response · Authors · 2024-11-24
> >
> > Thank you for your kind words. We will definitely implement all the promised changes before publication.

---

### Official Review · Reviewer_VHBa · 2024-11-04

**Soundness:** 2
**Presentation:** 3
**Contribution:** 2
**Rating:** 3
**Confidence:** 4

**Summary:**

The authors propose a method to speed up the benchmark evaluation for large language models. The method is to fit a neural process to existing benchmarks results and to predict a larger set of evaluation results from a smaller set of results. In doing so, the authors hope to leverage the correlations between the results of a language model on different prompts in a benchmark. What works best in the empirical evaluation is a method that chooses prompts based on a reinforcement learning heuristic.

**Strengths:**

(1)

The general research direction is well motivated. Benchmark evaluation for LLMs is indeed a costly enterprise. Methods to compare models from fewer examples could potentially be very useful.

(2)

The authors tried out several different model fitting methods. The RL method required some clever reward engineering. The amount of work displayed here is adequate for an ICLR paper.

**Weaknesses:**

(1)

There is a major methodological weakness in the evaluation.

The authors only test how well their method can fit existing benchmark results. To do so they split the the benchmark results into training and test, fit the method on train, and evaluate the approximation errors on test. You might call this the missing value interpolation error on the benchmark. This quantity, however, does *not* shed light on what would happen if a new model entered the benchmark. The new model might be out-of-distribution with respect to existing benchmark results. This can easily happen if the model answers prompts in a manner that's significantly different from prior models, as is the case with major model improvements. In this case, the interpolation error on existing benchmarks does not and cannot give an indication of how well the proposed method would do on the new model.

To strengthen the evaluation part of this paper, it would be helpful to train the method on a subset of older models and then test what would happen if newer models entered the benchmark. Does the method correctly identify the relative odering of the new models? This is a more meaningful criterion than the missing value interpolation error on the entire leaderboard.

(2)

The method is highly complex with a lot of moving pieces and knobs to turn. In particular, there are several different confounding sources of error: the prompt embedding, the selection method, the neural approximation. This leads to several problems of which I'll mention two:

(a) It's somewhat unclear how a benchmark designer is supposed to implement this method without already having a large amount of benchmark results to draw on. In particular, it's not clear how to get this method off the ground for a new benchmark.

(b) As the set of models in the benchmark grows, it would seem that this method requires retraining on the growing set of data points. This is an issue as it would possibly reorder the existing model ranking.

(c) The primary purpose of a benchmark is to standardize an evaluation protocol. With so many moving pieces and design choices, a benchmark based on this method will likely be harder to standardize. Moreover, the many design choices will make the benchmark appear arbitrary and reduce trust.

(3)

Given that the method necessarily introduces bias, it would be good to combine it with existing bias mitigation methods such as prediction powered inference. See, e.g., this reference and prior work: https://arxiv.org/abs/2403.07008

**Questions:**

It's not clear to me how to apply this method:

(a) At the design stage when not benchmark data is known.

(b) When new models are introduced to an existing benchmarks.

Could you please explain how this method could be incorporate into the design of benchmarks from the get go? What do you do right at the beginning?

Moreover, as the set of models in a benchmark grows, how does the method keep up with the new data? Do you recommend retraining after each model is introduced?

Finally, how does the computational complexity of implementing your method (possibly with retraining after reach new model) compare with simply evaluating on more prompts?

---

> ### Author Response · Authors · 2024-11-14
>
> We deeply appreciate you taking the time to provide such detailed and insightful feedback. We believe that by addressing these concerns, we can significantly strengthen the contributions and practical impact of our work. We hope that with these improvements, you would reconsider your rating and advocate for the publication of our paper.
>
> **1. Evaluation on New Models (Out-of-Distribution)**
>
> You are correct that evaluating the interpolation error on existing benchmark results may not fully capture the performance on new models that could potentially be out-of-distribution compared to the models used for training. This is a valid concern, and we have already considered it in our evaluation methodology.
>
> Note that our data splitting strategy (lines 360-371) is based on models, not prompts. Our train and test sets contain non-overlapping sets of models to assess how our methods generalize to new models. Due to space limitations, the data split details are moved to Appendix E. We will move them to the main text for the camera-ready version.
>
> For the OpenLLM Leaderboard and MMLU benchmarks, we leveraged the natural split based on the submission dates of the models, dividing them into older and newer groups accordingly. We evaluate our method's performance in accurately estimating the benchmark scores for the newer group of models after training on the older group. This evaluation setup directly assesses our approach's ability to generalize to unseen, potentially out-of-distribution models.
>
> For benchmarks like HELM-Lite and AlpacaEval, where submission dates were not available, we conduct ablation studies by dividing the models based on their organizations. For HELM-Lite, we use proprietary models, such as GPT-4 and Claude, for training and test on open-source models, such as LLaMA and Mistral. For AlpacaEval, we do the opposite, using open-source models for training and proprietary models for testing. Please see details in lines 447-464 and results in Figure 2.
>
> Overall, our RL-based acquisition policy generalizes better to the out-of-distribution (OOD) models compared to other baselines. We appreciate you highlighting this important aspect of our work, and we will ensure that our evaluation methodology and results are clearly described in the paper to address your concern.
>
> **2. Ablation Studies on Moving Pieces**
>
> You're correct that our method involves several components, including prompt embeddings, subset selection policies, and neural process approximation, which could potentially introduce confounding sources of error. To address the potential confounding sources of error, we have performed the following ablation studies:
>
> 1. Prompt Embeddings: We have evaluated the impact of using different prompt embedding models, ranging from large pre-trained models like SFR and E5 to smaller models like BGE-large and BGE-small. Our results, presented in Table 3, demonstrate that our method's performance generally improves with more powerful embedding models that can better distinguish text inputs. Based on these findings, we recommend using the most powerful embedding model available for the best performance.
>
> 2. Subset Selection Policies: We have extensively studied and compared various subset selection policies, including random policies (uniform sampling and stratified random sampling), static policies (clustering-based on embeddings, scores, and IRT features), and our proposed dynamic RL-based policy. Our empirical results, shown in Figure 1, clearly indicate that the RL-based policy outperforms other baselines across multiple benchmarks, demonstrating its effectiveness in selecting informative prompts.
>
> 3. Neural Process Approximation: To assess the impact of the neural process approximation component, we have compared our proposed method, which leverages the neural process to predict evaluation scores for unselected prompts, with a baseline approach that directly aggregates the acquired evaluation scores without any prediction. The results, presented in Table 2, show that the neural process generally provides better benchmark performance estimation, highlighting the importance of this component in our method.  Furthermore, In Table 1, we compare our neural process model and the IRT model proposed in tinybenchmarks, using the prompt selections from our RL-based policy with those from tinybenchmarks. The results demonstrate that for both prompt selections, using our neural process produces better benchmark performance estimates, indicating that our model better captures the dependencies and predicts the missing evaluation scores more accurately.

---

> > ### Author Response · Authors · 2024-11-14
> >
> > **5. Potential Impact on Benchmark Standardization and Trust**
> >
> > You raise a fair point regarding the potential challenges in standardizing a benchmark based on our method, given the multiple design choices and moving pieces involved. However, we would like to emphasize that the primary goal of our work is to improve the efficiency and reduce the computational cost of LLM evaluation, while maintaining reasonable accuracy in performance estimation.
> >
> > As our results demonstrate, our method can achieve benchmark performance estimation within a reasonable error range (e.g., within 2% for most benchmarks), while significantly reducing the number of prompts required for evaluation. This computational efficiency is crucial for enabling frequent evaluations during model development and extensive hyperparameter tuning during inference, which are often hindered by the substantial costs associated with comprehensive LLM benchmarks.
> >
> > We believe that our method's potential for significant computational savings and efficient LLM evaluation outweighs the challenges associated with standardization. One potential mitigation is to use our efficient methods to select a subset of models of interest and run full evaluation only on those selected models.
> >
> > By leveraging our approach to identify the most promising models based on the efficient evaluations, we can then allocate resources to conduct comprehensive evaluations on this curated subset. This two-stage process could strike a balance between computational efficiency and standardization, as the final comprehensive evaluations would adhere to the existing standardized protocols.
> >
> > **6. Computational Complexity Compared to Full Evaluation**
> >
> > Regarding the computational complexity of our method compared to full evaluation, there are two main factors to consider:
> >
> > 1) The cost of acquiring evaluation scores for the selected subset of prompts, which is a fraction of the full evaluation cost.
> > 2) The cost of running the neural process model and acquisition policies.
> >
> > While the second factor introduces some additional overhead, the primary computational savings come from the reduced number of prompts that need to be evaluated. In our experiments, we observed that our method could achieve accurate performance estimation while acquiring evaluation scores for only a small subset (e.g., <1% for MMLU and OpenLLM, 1.5% for HELM-Lite) of the total prompts.
> >
> > The bottleneck of evaluation cost is on LLM inference, which typically requires high-end computation resources, while our neural process model and acquisition policy are much smaller (2-3 linear layers in our experiments), which can be executed efficiently.
> >
> > By significantly reducing the number of prompts that require LLM inference, our method offers substantial computational savings compared to full evaluation, even when accounting for the overhead of running the neural process model and acquisition policies. Additionally, the potential need for periodic retraining (which is only needed when new models are significantly different from existing ones) as new models are added incurs a relatively minor cost, as our models are lightweight and can be efficiently trained on the available evaluation data.
> >
> > **7. Combination with Bias Mitigation Methods**
> >
> > You make an excellent suggestion regarding combining our method with existing bias mitigation techniques, such as prediction-powered inference (Puri et al., 2023). We agree that this could be a valuable direction to explore, as our method may introduce biases due to the selective evaluation process. We will investigate the integration of our approach with bias mitigation methods for future works.
> >
> >
> > Thank you again for raising these important points and concerns. If any concerns remain unaddressed, please do not hesitate to provide further feedback, as we are committed to ensuring the robustness and reliability of our method.

---

> > > ### Comment · Reviewer_VHBa · 2024-12-03
> > >
> > > Thank you for your extensive responses, which I read with interest. I'm appreciative of the effort you put into this work and also your response.
> > >
> > > I re-read your paper in light of your comments. Here is where I landed:
> > >
> > > I paid closer attention to the performance of random sampling in the experimental evaluation this time around. The reason is that uniform sampling solves all the issues I pointed out: cold start, design stage, complexity, trust, uncertainty quantification, etc.
> > >
> > > It looks to me that uniform sampling is surprisingly competitive compared with much more complex policies. It achieves small error across the board while permitting statistical analysis. It's also easily interpretable by benchmark users and does not raise any trust issues. By definition, it's quite efficient, too. And it applies at design stage.
> > >
> > > So, the first rhetorical lift that any paper on this topic would have to do is to argue why random sampling isn't *good enough*. But the crux is I just wasn't persuaded that this is so.
> > >
> > > Even if on some benchmarks more complex methods have slightly smaller error, it is not clear me that this gain justifies the cost.
> > > We're going from 0.052 to 0.034 (Chatbot arena) or 0.018 to 0.013 (MMLU). But this improvement of 0.005 comes at the cost of complete loss of uncertainty quantification, loss of interpretability, massive increase in computational complexity, the cold start problem etc.
> > >
> > > It is not even clear to me that this 0.005 improvement is real in the sense that the error might be below the inherent noise level of the benchmark. We simply may not want to distinguish between models that differ by 0.005 on MMLU. We're probably better off calling it even at that point.
> > >
> > > I still see merit in your paper by demonstrating how even highly complex methods only provide modest gains over random sampling. But then the story shouldn't be: Use this complicated method. But rather:  `np.random.choice` is all you need.

---

> ### Author Response · Authors · 2024-11-14
>
> **3. Apply the Method at the Design Stage**
>
> You're correct that it is unclear how a benchmark designer would implement our method without already having a large amount of benchmark results to draw on. However, this is not the primary setting we target with our approach.
>
> Our method is designed to enable efficient evaluation of large language models on existing, well-established benchmarks that have already accumulated a substantial amount of evaluation data from various models. In such scenarios, our approach can leverage this prior evaluation data to learn the dependencies across prompts and develop effective acquisition policies, thereby reducing the computational cost of evaluating new models on the benchmark.
>
> That being said, if one were to apply our method to a completely new benchmark with no prior evaluation data, we acknowledge that it would be challenging to get the method "off the ground." In such cases, we propose a two-stage approach:
>
> Stage 1 (Cold Start): During the initial design phase, when no evaluation data is available, we can rely on simpler acquisition policies like random sampling or clustering based on prompt embeddings. These policies do not require any prior evaluation data and can be used to acquire an initial set of evaluation scores for a small number of seed models. As for the benchmark performance estimation, one can directly estimate it based on the scores on the selected prompts (corresponds to the "w/o pred" setting in Table 2). This will result in higher estimation error, but depending on the budget one can allocate, it provides a reasonable starting point.
>
> Stage 2 (Warm Start): Once we have acquired some initial evaluation data from Stage 1, we can then train our neural process model and RL-based acquisition policy on this seed data. From this point onwards, our method can be used to actively acquire evaluation scores for new models more efficiently, leveraging the learned dependencies and acquisition policies. Note that both our neural process model and acquisition policy can be trained with partial available scores, thus the collected scores in Stage 1 can be effectively used.
>
> We will include a detailed discussion of this two-stage approach in the paper. While our method may not be immediately applicable to brand-new benchmarks without any prior data, the two-stage approach offers a practical solution to bootstrap the process and subsequently benefit from the computational efficiency of our method as more evaluation data becomes available.
>
> **4. Handling New Models in an Existing Benchmark**
>
> As our results indicate, our approach can reasonably generalize to new models, even those that may exhibit different behavior compared to the models used for training. This generalization capability is demonstrated in our experiments where we split the models into older and newer groups (based on submission dates), train our method on the older group, and evaluate its performance on the newer, unseen group of models. We also tested generalization capability by dividing the models based on organizations (See Fig 2).
>
> However, we recognize that if the new models introduced to the benchmark exhibit behavior that is significantly different from all previously seen models, it is possible that our method may give higher estimation errors. Dealing with such out-of-distribution scenarios is an interesting and important direction for future work, not only for our method but also for the broader machine learning community.
>
> If significant distribution shifts are observed as new models are added to the benchmark, one potential approach could be to periodically retrain our method, incorporating the evaluation data from the new models. This would help our method adapt and capture the evolving dependencies across prompts and model behaviors.
>
> Alternatively, we could explore techniques from continual learning or lifelong learning, which aim to enable models to adapt to new data while preserving the knowledge acquired from previous data. By incorporating such techniques, our method may be able to update its understanding of prompt dependencies and acquisition policies in an incremental manner, without the need for complete retraining from scratch.
>
> While addressing out-of-distribution scenarios and distribution shifts is an important area for future research, we believe that our current work provides a solid foundation and demonstrates the potential of our approach in enabling efficient LLM evaluation within the scope of the existing benchmark data. We will include a discussion of these limitations and future directions in the paper, highlighting the need for further research in this area.

---

### Official Review · Reviewer_PgQH · 2024-11-05

**Soundness:** 2
**Presentation:** 2
**Contribution:** 2
**Rating:** 3
**Confidence:** 3

**Summary:**

This paper proposes the Adaptive Evaluation Acquisition (AEA) approach which is basically an adaptive subset selection of prompts for a model so that the generalization error on remaining prompts is low. Note that this is different than Active learning as the goal here is to perform better at evaluation time than training an accurate model. Here, I also want to point out that even though the authors have noted their setting is different than Active learning, their prompt selection strategy is basically an uncertainty sampling (max-entropy) approach (see eq (5)). Their key idea is to capture the dependencies between the prompts by modeling the conditional distribution between unobserved label $Y^u$ and observed label $Y^o$ as a stochastic process as they outline in eq(2). Since the integration is over a high dimensional latent space, following a standard procedure they optimize the evidence lower bound (ELBO) following variational autoencoder (VAE). Once the underlying modeling as a stochastic random process is fixed, they then use an active (or static) acquisition policy to choose the $Y^o$ in algorithm 1. They propose several acquisition strategies for selecting the prompts, ranging from static combinatorial optimization approach to max entropy. They evaluate this approach on several LLM dataset benchmarks.

**Strengths:**

1. The paper is well written as it clearly lays down the motivation of the paper, prompt subset selection for evaluation. However, how significant is the motivation is not clear to me.
2. The key idea is to capture the dependencies between the prompts by modeling the conditional distribution between unobserved label and observed labels using a stochastic process makes sense to me. This is standard VAE stuff.
3. They clearly explain the acquisition function used to select the prompts. However, these acquisition functions are well-known in active learning. So I cannot consider them a novel contribution of this paper.

**Weaknesses:**

1) The writing needs more improvement. For example in lines 86-87, the subset $o^*$ has cardinality $K$ but is indexed from $\{1, \ldots, N\}$. Is K < N Do you mean to say that there are some repetitive prompts?
2) The approach seems very similar to few shot learning with LLMs, however they do not do any comparison with these works. See my question 1.
3) The significance of the motivation is not clear to me. Also, it is not clear to me how this subset of prompts on which the LLM is evaluated captures the hardness of the task. See my question 2, 3.
4) The difference with Active learning is not very clear. This needs more explanation See my question 4.

**Questions:**

1. I understand the paper is doing subset selection of prompts for evaluation so that it finds the most informative prompts so that if the model performs well in these prompts, then the model can be said to perform well on all prompts from the test set. This is related to the few-shot learning with LLMs [a, b, c, d] and the relevant papers that cite them. Note that these works use subset selection during training time whereas this paper does subset selection of prompts during evaluation time. However, under the condition that training and testing prompts come from the same underlying distribution, how can the two approaches be different (or can be reconciled)?
2. The significance of this setting is not clear to me. Most importantly, a more detailed discussion on related work will bring this out. What are the works that are closest to your work that do some sort of active testing? The only reference I can find is the Kossen et al. 2021 (line 318).
3. I understand how the different acquisition functions capture prompt diversity and informativeness using embeddings. This is standard in AL/RL/bandit literature. However, an additional component in natural language generation/text generation is the difficulty of the task (such as chain-of-thought promoting or mathematical reasoning). How does your acquisition function take these into account?
4. Again, my final question is more in relation to Active learning. The approaches that you mention such as clustering, uncertainty sampling, and information gain are already well known. The combinatorial approach can be mapped to combinatorial semi-bandits (see [e]). In light of this can the authors shed more light on the key technical novelty/approach of this paper? Can I simply say that this work is an extension of active training (read active learning where you may or may not train the model and simply do few shot learning) to active testing for LLMs?

[a] Yiming Zhang, Shi Feng, and Chenhao Tan. Active example selection for in-context learning
[b] Yuanhan Zhang, Kaiyang Zhou, and Ziwei Liu. What makes good examples for visual in-context learning?
[c] Zhuosheng Zhang, Aston Zhang, Mu Li, and Alex Smola. Automatic chain of thought prompting in large language models.
[d] Subhojyoti Mukherjee, Ge Liu, Aniket Deshmukh, Anusha Lalitha, Yifei Ma, Branislav Kveton Experimental Design for Active Transductive Inference in Large Language Models
[e] Branislav Kveton, Zheng Wen, Azin Ashkan, Csaba Szepesvari, Tight Regret Bounds for Stochastic Combinatorial Semi-Bandits

---

> ### Author Response · Authors · 2024-11-14
>
> Thank you for the detailed review and insightful questions. We will do our best to address the issues and questions raised. We believe this is an important setting for the LLM evaluation community, as it can significantly reduce the evaluation cost, which is a critical challenge given the rapidly increasing scale and complexity of LLM benchmarks. We would greatly appreciate if you could advocate for publication if we have adequately addressed your questions and concerns.
>
> **1. Comparison with few-shot learning:**
>
> You raise a valid point about the connection between our work and few-shot learning with LLMs. While there are similarities in the sense that both involve selecting a subset of examples, the key difference lies in the objective. Few-shot learning aims to select examples that can effectively adapt a pre-trained LLM to a new task or domain, with the goal of improving the model's performance on that specific task. In contrast, our work focuses on selecting a subset of prompts for efficiently evaluating an already-trained LLM across a diverse set of tasks or capabilities, without any further fine-tuning or adaptation. The objective is to accurately estimate the model's overall performance on the entire benchmark while minimizing the number of prompts required for evaluation.
>
> Despite this difference in objectives, we acknowledge that there could be potential synergies between the two lines of work, and a more detailed discussion comparing and contrasting them would be beneficial. We will incorporate a dedicated section in the related work to establish this connection and highlight the key distinctions.
>
> **2. Significance of the motivation:**
>
> The motivation for our work stems from the increasing complexity and comprehensiveness of LLM evaluation benchmarks, which often comprise thousands or even millions of prompts spanning diverse tasks and capabilities. Evaluating an LLM on such large-scale benchmarks can incur substantial computational costs, both in terms of resources and time. For instance, as mentioned in the paper, evaluating on the HELM benchmark requires 4,200 GPU hours for a 176B model and over \$9,000 in API costs. These substantial costs can hinder frequent evaluations during model development and limit extensive hyperparameter tuning during inference.
>
> Our work aims to alleviate these challenges by reducing the number of prompts required for accurate LLM evaluation, thereby lowering the computational burden and associated costs. This is particularly important in scenarios where frequent evaluations are necessary, such as during model development or when exploring various decoding and prompting strategies.
>
> We will strengthen the motivation section in the paper to better highlight the practical significance and potential impact of our work in enabling more efficient and cost-effective LLM evaluation.
>
> **3. Capturing task hardness in the acquisition function:**
>
> You make a good point. The current LLM evaluation benchmarks primarily focus on assessing the final answer or output quality, rather than explicitly evaluating the underlying reasoning capabilities or the solution process followed by the model.
>
> While the reasoning requirements for different prompts may vary in difficulty, such as those involving mathematical problem-solving or multi-step logical reasoning, these complexities are effectively integrated into the final evaluation score assigned to each prompt. In other words, a model with stronger reasoning capabilities is likely to perform better and achieve higher scores on prompts that demand complex reasoning skills, as compared to models with more limited reasoning abilities. Consequently, the evaluation scores themselves implicitly capture the model's proficiency in handling prompts with varying levels of reasoning difficulty.
>
> Our acquisition functions, by leveraging the dependencies among these evaluation scores, indirectly account for the differences in reasoning requirements across prompts. Models that excel at complex reasoning will have higher scores on the corresponding prompts, and our methods will prioritize the selection of such informative prompts during the acquisition process.
>
> While our current approach does not explicitly model or quantify the specific reasoning types or difficulty levels, it effectively captures these nuances through the evaluation scores, which serve as a holistic measure of the model's performance on each prompt, integrating all aspects of the required reasoning capabilities. However, we acknowledge the value of your suggestion and the potential benefits of explicitly modeling and considering reasoning complexity during the acquisition process. As future work, we will explore techniques to incorporate more fine-grained representations of reasoning requirements. This could enable a more nuanced analysis and targeted evaluation of specific reasoning skills, in addition to the overall performance assessment provided by the existing benchmarks.

---

> ### Author Response · Authors · 2024-11-14
>
> **4. Technical novelty and comparison with active learning:**
>
> While our work shares similarities with active learning in the sense that we aim to select the most informative examples, there are key distinctions in terms of the objectives and settings.
>
> In active learning, the goal is to select a subset of examples to label and then use them to train a better model. The objective is to improve the model's performance on the task at hand by acquiring useful training examples. In contrast, our work focuses on selecting a subset of prompts for efficient evaluation of an already-trained LLM, without any further fine-tuning or adaptation. The objective is to accurately estimate the model's overall performance on a comprehensive benchmark while minimizing the number of prompts required for evaluation.
>
> Additionally, in active learning, the labeled examples are directly used for training, and the model's performance is evaluated on a separate test set. In our setting, the acquired evaluation scores are used to predict the scores for the remaining prompts, and the final performance estimate is derived from both the acquired and predicted scores.
>
> While we do leverage some techniques from active learning, such as uncertainty sampling and information gain, our primary contribution lies in adapting these methods to the specific context of LLM evaluation and developing novel acquisition policies tailored to this setting. For instance, our RL-based acquisition policy (see equation 7) incorporates auxiliary information from the generative model and intermediate rewards to guide the exploration of the prompt space more effectively. Furthermore, we introduce the cold-start problem, where new prompts are added to the benchmark without any prior evaluation scores, and propose techniques to handle this scenario.
>
> We will clarify the distinctions between our work and active learning in the paper and provide a more detailed comparison to highlight our specific technical contributions in the context of efficient LLM evaluation.
>
> **5. Others:**
>
> > The writing needs more improvement. For example in lines 86-87, the subset $o^*$ has cardinality $K$ but is indexed from $1, \ldots, N$. Is $K < N$ Do you mean to say that there are some repetitive prompts?
>
> The optimal subset $o^*$ is a subset of $\{1, . . . , N\}$ with cardinality $K$, where $K \leq N$. There are no repetitive prompts; instead, we select $K$ unique prompts from the set of $N$ prompts in the benchmark.
>
>
> Thank you once again for the insightful review and constructive feedback. We appreciate the opportunity to clarify and address the raised issues, which will undoubtedly strengthen the paper's clarity and positioning. If you have any further suggestions or comments, please let us know.

---

> > ### Comment · Reviewer_PgQH · 2024-12-03
> > **Response to authors**
> >
> > I am highly unconvinced with the answers of the authors. Here is my take:
> > 1. Yes, traditional AL deals with training a model on an informative and diverse subset selection of examples (read prompts). However, as I have shown, there are plenty of examples where they do not need to train the model and just do few shot learning with in-context examples. Here is one more paper as a read-up on the associated works [f].
> > 2. Motivation somewhat makes sense.
> > 3. The answer to the task hardness is unsatisfactory. Sentences like *"Our acquisition functions, by leveraging the dependencies among these evaluation scores, indirectly account for the differences in reasoning requirements across prompts"* is very vague. This needs to be shown concretely through experiments and evaluations.
> > 4. Again, you need not retrain the model, yet use AL techniques to balance uncertainty and diversity sampling. [f] is an example where they do not retrain the model. They are similar papers as well, do a literature search.
> >
> > [f] Optimal Design for Adaptive In-Context Prompt Tuning in Large Language Models

---

### Meta-Review · Area_Chair_WMRR · 2024-12-19

**Metareview:**

The paper studies strategies to improve evaluation efficiency by actively selecting a subset of prompts to evaluate and then predicting the performance of the rest of the prompts. The paper empirically demonstrates that the proposed approach can reduce the number of evaluation prompts.

The weaknesses pointed out by reviewers include its comparison to related work on few-shot learning in LLM and existing techniques in active learning and active testing, the complexity of the proposed approach, and its advantages over the very simple baseline -- random sampling.

**Additional Comments On Reviewer Discussion:**

While the reviewers acknowledged the authors' effort during the rebuttal time, there are still some concerns left. In particular, reviewers are still unsatisfied with the discussion regarding comparisons to related work on active learning and active testing. Another reviewer pointed out that the comparison to uniform sampling is not convincing: while it is true that the proposed approach achieves slightly better performance in the experiments, the uniform sampling strategy is also competitive. Thus, the authors are encouraged to explain why their approach should be preferred over uniform sampling, especially considering the simplicity of the uniform sampling strategy.

---

### Decision · Program_Chairs · 2025-01-22

Reject